# Assessment of a Fully Renewable Generation System with Storage to Cost-Effectively Cover the Electricity Demand of Standalone Grids: The Case of the Canary Archipelago by 2040

**Yago Rivera-Durán [1], César Berna-Escriche [1,2,\*], Yaisel Córdova-Chávez [1] and José Luis Muñoz-Cobo [1]**

[1] Instituto Universitario de Investigación en Ingeniería Energética IIE), Universitat Politècnica de València (UPV), Camino de Vera 14, 46022 Valencia, Spain

[2] Departamento de Estadística, Investigación Operativa Aplicadas y Calidad, Universitat Politècnica de València (UPV), Camino de Vera 14, 46022 Valencia, Spain

\* Correspondence: ceberes@iie.upv.es; Tel.: +34-963879245

**Abstract:** The change towards a clean electric generation system is essential to achieve the economy decarbonization goal. The Canary Islands Archipelago confronts social, environmental, and economic challenges to overcome the profound change from a fossil fuel-dependent economy to a fully sustainable renewable economy. This document analyzes a scenario with a totally renewable generation system and with total electrification of the economy for the Canary Islands by 2040. In addition, it also shows the significant reduction in this fully renewable system when an optimized interconnection among islands is considered. This scenario consists of a solar PV system of 11 GWp, a wind system of only 0.39 GWp, a pumped storage system of 16.64 GWh (2065 MW), and a lithium-ion battery system of 34.672 GWh (3500 MW), having a system LCOE of 10.1 cEUR/kWh. These results show the certainty of being able to use an autonomous, reliable, and fully renewable system to generate and store the energy needed to dispense with fossil fuels, thus, resulting in a system free of greenhouse gas emissions in the electricity market. In addition, the proposed system has low energy wastage (less than 20%) for a fully renewable, stand-alone, and off-grid system.

**Keywords:** renewable energy; reversible pumping storage; mega-batteries; standalone system; grid interconnection; exploratory analysis; sensitivity analysis

## 1. Introduction

According to the IEA report "World Energy Outlook 2020" [1], world energy demand has continuously increased over the last decades, except for a slight decrease in 2020 because of the COVID-19 pandemic. The growing tendency has returned in 2021 even though the COVID-19 pandemic has not yet ended [2]. Most of this produced energy comes from fossil fuels, and similar figures are shown when regarding power generation. In this case, approximately two-thirds are generated through fossil fuels [3].

This situation of generation based on fossil fuels is unsustainable, fundamentally posing, on the one hand, a more than foreseeable depletion of fossil fuels in the medium term if the current rate of consumption is maintained [4,5]. The second problem focuses on pollutant emissions, particularly greenhouse gas emissions, which are gases produced in large quantities when energy uses are centered on fossil fuels as combustibles or raw materials [6,7].

Therefore, there are many reasons for renewable energies to be present or even to be the only generation sources used, aiming to reduce or eliminate fossil fuels [8]. Concentrating on electricity generation, the use of renewable energies is a must, since otherwise it is impossible to reach the ambitious objectives of $CO_2$ emissions reduction, combined with the sharp increase in the use of electricity, which has the final energy consumption

of all countries [9], expecting to exceed 30% of the total amount in a short time in many of them [10].

The described problem is further complicated in remote regions, such as islands [11]. This is because their small size and inaccessible location make connecting to a large grid difficult or even technically or economically impossible. Therefore, in the case of islands, the typical solution is to have a fossil fuels-based energy system (coal, gas, and diesel), since these systems have the advantage of high reliability. However, as disadvantages, these systems have a huge emissions problem and also a strong dependency on a big and complex supply chain. In addition, the countries producing these fuels are in many cases unstable, with the consequent risk of shortages that can reduce system reliability, added to the inconvenience that the prices of these fuels suffer large oscillations with frequently and unexpected rising of prices due to cartel decisions [11].

Consequently, from both environmental and strategic (energy autonomy) points of view, renewable energies are an option that will be implemented in many places [12–15]. However, when renewable energies take on an important weight in the generation of power, a series of challenges arise, mainly associated with the intrinsic variability that this type of source presents [16–18]. Specifically, the two renewable sources that are currently capable of covering the existing energy needs, solar photovoltaic and wind power, present a strong variability, with longer or shorter periods of low or strong wind, solar cycles, cloudy or rainy days, etc. All this means that to cover the energy needs with these sources, the system needs to be oversized and must include large storage systems to be able to absorb at least part of the excess to have them available in case of need. Even so, there will inevitably be excesses, but in this case, they will be more limited. Thus, the optimal generation and storage should, therefore, be sized, usually from an economic point of view, i.e., installing a generation power and storage capacity able to cover the energy needs at the lowest cost. Storage systems (pumping stations, mega-batteries) and large-scale generation (wind and solar PV) present an added problem in many cases on islands, caused by the scarcity of appropriate sites that would be required [19].

Focusing specifically on the case of the Canary Islands, this archipelago consists of seven islands separated by less than 100 km from the Moroccan coast at its closest point and about 300 km at its furthest one, being about 1500 km from the mainland European coast. All 7 islands are inhabited by more than 2 million people [20], and the 2 major islands, Gran Canaria and Tenerife, have around 950 and 850 thousand inhabitants, respectively. Lanzarote and Fuerteventura have approximately 150 and 120 thousand, respectively; finally, La Palma, La Gomera, and El Hierro have a population of slightly above 80, 20, and 10 thousand inhabitants, respectively (official data of 2020). The total final energy demand for the whole archipelago in the year 2019 was about 9.4 TWh (values which had remained almost constant during around the last 10 years and that have been reduced in 2020 and 2021 mainly because of the COVID-19 pandemic) [20]. Only slightly below 1.5 TWh of the total energy demand were generated through renewable sources, i.e., around 16% of the total generation. The energy demand broken down by islands and expressed in GWh per year (also expressed in percentages in brackets) is 3711 (40%), 3582 (38.4%), 906 (9.7%), 717 (7.7%), 281 (2.9%), 76.9 (0.8%), and 62.4 (0.5%) for Tenerife, Grand Canary, Lanzarote, Fuerteventura, La Palma, La Gomera, and El Hierro, respectively.

This study proposes an autonomous generation system of electric power based on renewable energy and storage technologies (reversal pumping and mega-battery systems) with no $CO_2$ emissions. The optimizing criteria are based mainly on economic reasons, but energy waste optimization and land occupation have also been considered. The analysis has been applied to the islands aiming to be energetically autonomous, estimations for each island have been carried out, and the comparison to a future scenario with the electric system interconnected between them has been made.

In view of the aforementioned situation, renewable energy generation systems have to be based on solar photovoltaic and wind technologies, because are the uniquely mature-enough technologies that currently might substitute conventional fossil technologies

[8,21]. This is because other mature technologies that could also help in this process, such as hydroelectric power stations, are not available due to the absence of water resources in the Canary Islands. The solution to these feasibility problems, in all likelihood, involves the installation of high-energy storage capacities [22,23]. Consequently, the combined use of a renewable energy system backed by a high-capacity storage system will satisfy the local energy demand much more efficiently than a stand-alone renewable energy facility, due to the inherent variability of renewable sources [24].

All countries of the European Union must face the decarbonization of the economy by 2050. In this sense, the Canary Islands are working with a clear time limit in their strategy to reduce fossil fuels dependence and, particularly in the case of the Canary Islands, they have the advantage of the rich natural resources of the archipelago, such as the wind and the sun. The Canary Islands Government and the Spanish national government are yet more optimistic and have scheduled to bring forward the end of the decarbonization process by 10 years (PTECan project) [25,26].

As a vanguard of this ambitious project, it is worth mentioning that in recent years all of the Canary Islands have been expanding their wind farms and solar parks to a greater extent. In addition to the projects that are already in the planning or execution phase, many other efforts are necessary to achieve the ambitious results that are being sought. For instance, the Chira-Soria pumping reversal station project is being carried out on the Grand Canary Island [27], a facility that involves a high storage capacity, between 3.2 and 3.6 GWh, and a total generation capacity of 200 MW, with an estimated cost of around EUR 400 million. Between Lanzarote and Fuerteventura (Playa Blanca–La Oliva), there currently exists a submarine interconnection of approximate 16 km, 132 kV, and 121 MVA of transport capacity [28], while there is a project for another interconnection between Tenerife and the La Gomera islands (Chío-El Palmar) of 42 km at 66 kV with a capacity of 50 MVA, with a planned cost of EUR 103 million [29]. Consequently, many other projects are needed, whether for the installation of renewable generation sources, storage facilities, or interconnection grids between islands.

Various applications throughout the world have proven that electrical connections between islands are useful, since they provide several advantages, such as reliability, greater sustainability, cost reduction, and increased efficiency [30,31]. Examples of these implementations are in Greece, the Philippines, or Ecuador, and in Spain (between the peninsular territory and the Balearic Archipelago, as well as the abovementioned interconnection in the Canary Archipelago between Fuerteventura and Lanzarote). In fact, there are also some studies that analyze the Canary Islands themselves [30,32], but these usually deal with the interconnection between two islands and/or without making future estimates, although there are some studies that speak of the feasibility of interconnections between the islands of the archipelago [33,34]. This means less energy (forecasts of almost double the demand by 2040), with less variability (now generation is almost 90% based on fossil fuels, which allows for easy regulation), and only between two islands (approximately 75% of the demand but with the simplest grid, more concentrated generation, and storage, etc.).

To close this preliminary part, it can be pointed out that the research tackles the goal of a zero-emission generation system based entirely on renewable energies, meeting both technological and economic criteria, and achieving 100% demand coverage, given the need for high system reliability in the case of an autonomous network. This study addresses the energy autonomy of an archipelago of considerable size, which is in a relatively dispersed layout with seven different islands. Population and energy demand has also been considered with the added difficulty of considering long-term future demand forecasts. The islands have a total surface area of 7472 km², will have around 2,500,000 inhabitants by 2040, and an electricity demand estimated at up to 16–18 TWh per year also by the year 2040 [25].

To address the present study, we used the software developed by the National Renewable Energy Laboratory (NREL), called the Hybrid Optimization of Multiple Energy

Resources (HOMER) [35]. The method followed by the software is mainly economic, since the code estimates the optimal size of a system based on the investment to be made, the LCOE (levelized cost of energy), and the amortization based on the energy sources to be installed [35]. The HOMER software has been extensively tested in the simulation of hybrid renewable systems, with several generation sources and storage technologies all around the world [36,37]. These research works include some studies related to the Canary Archipelago, but all deal with problems of a minor nature, such as for a single island, or at most for the two largest islands [21,30,38].

In order to complete the previous objectives, Section 2 describes the current electricity supply of the Canary Archipelago. Section 3 focuses on the methodology developed to carry out the current analysis. The contextualization of the problem is discussed in Section 3, and the three scenarios under consideration are described. Section 4 below briefly describes the characteristics and the demanded information required for all the systems necessary to make the simulations within the desired horizon, both for generation and storage technologies. The principal results of the simulations conducted are presented in Section 5, with the corresponding analysis and discussion. Section 6 is dedicated to summarizing the conclusions of the current study in terms of the generation system in the three scenarios under consideration. Some possible future work is also mentioned in this section.

## 2. The Electricity Supply in the Archipelago

The Canary Archipelago is a middle-sized group of seven major islands. It has a total population of around 2.3 million people, and forecasts of more than 2.5 million by 2040. Table 1 includes general information on the islands and key data on the electricity supply. The islands have focused on the tourist sector (primarily for international travelers), and currently have a significant degree of maturity, meaning that the energy demand of all islands is highly stable. In fact, it has not changed considerably in the last 15 years; the only changes which have taken place from 2020 until today were because of the COVID-19 pandemic. The total number of tourists that visit the island is around 15 million people, and considering an average length of stay of approximately 1 or 2 weeks, this means that the population of the Canary Islands, on average, is increased by 10 to 20% during these times. However, the tourists' contribution is being implicitly considered in the presented energy data, since those numbers have stabilized during the last years.

**Table 1.** General data and power details of the Canary Archipelago. [20]

| | Tenerife | Gran Canaria | Lanzarote | Fuerteventura | La Palma | La Gomera | El Hierro | Total |
|---|---|---|---|---|---|---|---|---|
| Surface (km$^2$) | 2034 | 1560 | 845 | 1659 | 708 | 369 | 268 | 7472 |
| Population (thousand) | 917.8 | 851.2 | 152.29 | 116.89 | 82.67 | 21.50 | 10.97 | 2185 |
| Generation per Year (GWh) | 3711 | 3582 | 906.1 | 716.8 | 281.0 | 76.85 | 62.43 | 9336 |
| Fossil Fuel | 3015 | 3028 | 826.5 | 636.7 | 251.9 | 76.70 | 20.74 | 7855 |
| Renewable | 696.1 | 554.0 | 79.62 | 80.11 | 29.08 | 0.154 | 41.69 | 1481 |
| Installed Power (MW) | 1426 | 1224 | 264.1 | 227.6 | 117.1 | 21.6 | 37.7 | 3306 |
| Fossil Fuel | 1112 | 1024 | 232,3 | 187,0 | 105.34 | 21.17 | 14.91 | 2696 |
| Renewable | 314.5 | 199.9 | 31.79 | 40.57 | 11.80 | 0.37 | 22.83 | 609.5 |
| GHG Emissions (tCO$_2$) | $2.12 \times 10^6$ | $2.07 \times 10^6$ | $5.5 \times 10^5$ | $4.83 \times 10^5$ | $1.72 \times 10^5$ | $5.28 \times 10^4$ | $1.43 \times 10^4$ | $5.45 \times 10^6$ |

The electricity cost when disaggregated by sources is unavailable on the REE webpage; therefore, the assumed cost is the real hourly cost for the entire electricity mix of the Canary Archipelago (both no renewable and renewable plants included). The energy cost data for 2019 has been collected from the Spanish electric system operator [39]. The minimum, maximum, and mean hourly cost for 2019 was 10.4, 24.4, and 15.3 cEUR/kWh, respectively, while the costs for the year 2022 up to the 1st of October were 8.03, 21.0, and 56.2 cEUR/kWh, respectively. These two periods of time have been selected

since 2019 is a conservative year in which the energy cost stabilized, while the current year has well-known international uncertainties which lead to an important increase in electricity costs.

With regard to greenhouse gas emissions (GHG) in the Canary Islands, except in specific years, progress has been marked by sustained growth during the period 1990–2005. From 2006 onwards, the trend reversed, and negative annual growth began to be recorded. This change in trend continued until 2016 and 2017, years in which positive growth was recorded again, only to fall again in 2018 and 2019. Therefore, it can be said that since 2011, the GHG has been in a relatively narrow band of reduced variability, between approximately 13-14 × $10^6$ $tCO_2$-eq [20]. In numerical terms, GHG emissions in the Canary Islands in 2019 were 13.04 × $10^6$ $tCO_2$-eq. Of these 13 million tons emitted in 2019, almost 88% came from combustion activities (split almost 50/50 between activities related to the energy and transport sectors). Of the remaining 12%, around 8% came from waste treatment and disposal, while the remaining 4% of emissions came from other industrial processes and agriculture.

### 3. Methodology

In order to carry out the code estimations, a rigorous methodology has been followed, which includes a detailed introduction of the entered information required to execute the simulation and a scheme of the steps used, as shown in Figure 1. Among the necessary input data, the following can be mentioned: annual information on the energy demand to be included or, for future estimations, their forecasts; technical and cost information on the generation system to be considered (in the case of the current study, wind and photovoltaic power plants); technical information and cost of the storage system (mega-batteries and reversible pumping); the energy resources available for each generation system (wind and solar resource available at the selected sites); other additional economic data (for example, the annual interest rate and the useful life of the project). Based on all these data as input in the HOMER software, the most efficient combination of these generation systems can be estimated, so that all the energy demanded by the system is supplied. In particular, the nominal power to be required, the generation of power from each system, and the necessary storage capacity, etc., are determined. Economic information, such as LCOE, initial capital, net present cost (NPC), payback, and internal rate of return (IRR), are also provided by HOMER. The selection criteria in the methodology are the economical ones previously mentioned but keeping $CO_2$ gas emissions at zero.

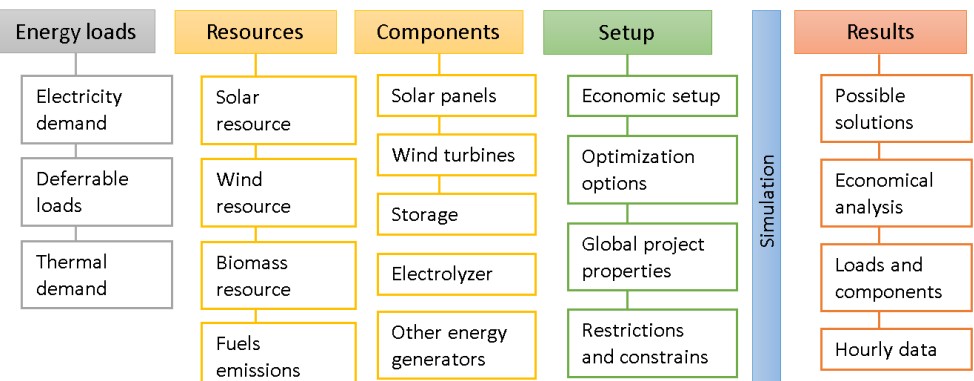

**Figure 1.** Schematic overview of the HOMER software information of inputs and outputs.

Economic criteria implicitly imply a trade-off between the sizing of the generation and storage facilities to satisfy the demand needs. Since the reached solution is the one in which the cost is the lowest, this largely means that the size of the system is as small as possible while also always covering the energy needs. In other words, the optimum point between the oversizing of the generation and storage systems is reached. To make these

estimates, economic data for the year 2019 have been used as the background, as it is assumed that the cost variation of the technologies used by 2040 will remain constant, which a priori should be true, since the technologies used are all at a fairly high level of maturity. The methodology has been tested in two distinct scenarios, both for the year 2040. These are coverage of the demand in each island separately and of an aggregated electric system for all of them, always considering the electrification of the economy.

The HOMER software performs simulations of the operation of each system analyzed through an energy balance at every time step defined in the code, the most common being the hourly balance, which is the one used in the current study. For this, it compares the energy demanded at each time step with that which can be distributed by the generation system under analysis, as well as how to operate the generators and whether it is required to make use of the batteries. Subsequent to simulating all system configurations, HOMER provides a list of feasible systems sorted by net present cost (NPC). The software displays a list of feasible solutions that meet all requirements. The global optimal solution tops the list, although other options can also be considered. For instance, other criteria can be considered, such as lower total installed power, minimization of energy wastages, the weighted combination of several of these factors, etc. In this case, as is usually carried out by other researchers [21,30,35,38,40], the global optimum has been considered as the better solution.

The uncertainties that must be considered to properly estimate future demands, in this case for the year 2040, will be associated with the evolution of electricity demand, electricity costs, wind/solar resources, and storage technologies. For this, different possible future scenarios have been contemplated and evaluated based on different aspects, such as the uncertainty in the evolution of demand, the degree of electrification of the economy, the degree of penetration of the electric vehicle, the possibility of implementing policies demand management, favoring self-consumption, etc. Finally, the optimal solution is provided for what is considered the most likely future scenario, which is fundamentally based on a work carried out recently focused on the Canary Archipelago itself [41] and on the Canary Islands Government [42–44]. To quantify the cost savings of electricity itself that would be achieved with this autonomous and totally renewable system, the current cost of electricity has been considered (that of the year 2019 has been taken, as it is a value not affected by the strong existing uncertainties that have multiplied the price by more than double), that is, it has been conservative, since it is not expected that the future cost of electricity will return to the figures considered, so the feasibility obtained from the alternative generation systems would still be elderly. The savings due to the elimination of greenhouse gas emissions have also been considered; for this the "cost per ton of $CO_2$" emitted into the atmosphere has been considered (also taken from the emission figures for the year 2019). Wind and solar radiation resources could also present uncertainties, but for the current analysis period of about 20 years, no substantial changes are assumed, and the typical values (from the Global Wind Atlas and PVGIS) are considered to be representative

### 3.1. Scenario of Autonomous Islands Generation

In this first scenario, the implementation of the necessary inputs for the software will be carried out for each of the archipelago islands separately and for the most probable scenario in the year 2040. On each island, the particular conditioning aspects for the different technologies are taken into account. As anticipated above, this scenario is based on the estimates carried out in different studies for the Canary Islands themselves; specifically, an in-depth analysis focused on the Gran Canaria Island is of particular interest [25,41,42]. This scenario consists of the total electrification of the economy with the implementation of strong efficiency measures and favoring self-consumption. Thus, in the end, translates into an increase of approximately 100% compared to the current values of electricity demand. Figure 2 shows the forecasted aggregated value of all the islands while considering that, by 2040, the percentage of consumption of each island will remain

approximately constant to the current one over these years, so it is possible to obtain that percentage for each of them.

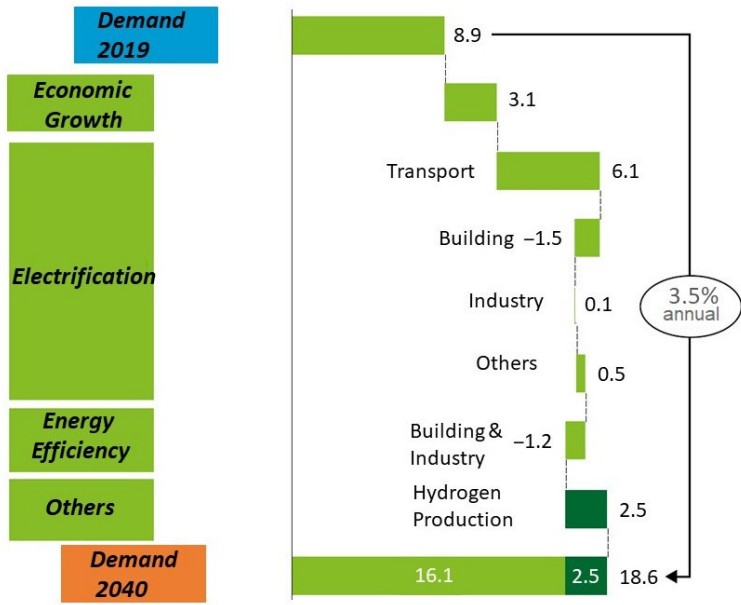

**Figure 2.** Forecast of the energy demand (TWh) in the Canary Archipelago from 2019 to 2040 (based on *[25]*).

The estimates of each island have been carried out by considering that the average hourly demand will have the same shape as the current one, with shape changes only being caused by the distribution resulting from the recharging of electric vehicles. Then, the contribution of electric vehicles has been estimated through the weighted ponderation of six profiles of electric vehicle recharge methods (homes and public roads, workplaces, hotel parking, shopping centers, and regular recharging points) for each island [42]. For the remaining demand, the current hourly demand profiles have been taken, but broken down by sector (residential, commercial, industrial, public administration, lodging and other uses) [43] and multiplied by the factor that contemplate the estimated increase until 2040. In other words, it has been considered that the current profiles and consumption proportions of each island are maintained, and a multiplying factor has been applied to include the economic growth until 2040. As an example, Figure 3 shows the shape of the estimated demand curve of an actual typical January day and for the year 2040 in this scenario of total electrification for the island of Lanzarote (for the remaining islands, a similar procedure has been carried out for the estimation, and the shape is also quite similar in all of them).

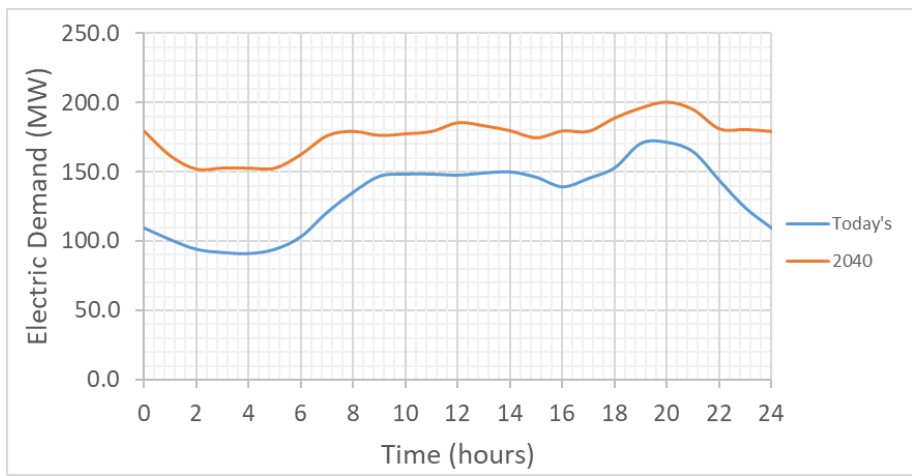

**Figure 3.** Current electricity demand profile and forecast of the island of Lanzarote for 2040.

### 3.2. Scenarios of Interconnected Grid between Islands

In the current case, the calculations are made for the aggregate system, but the most suitable locations for the different generation and storage systems have been sought. That is to say, for example, for the installation of the pumping systems, the appropriate locations existing on the different islands will be used, while for the wind turbines two locations have been sought close to the islands with the highest consumption so as not to have to oversize the interconnection between the two. In this scenario, the aggregate values shown in Figure 2 are used, as well as the aggregate demand curve. As an example, Figure 4 shows the aggregated curves for a typical January day for the current and forecasted 2040 demand values.

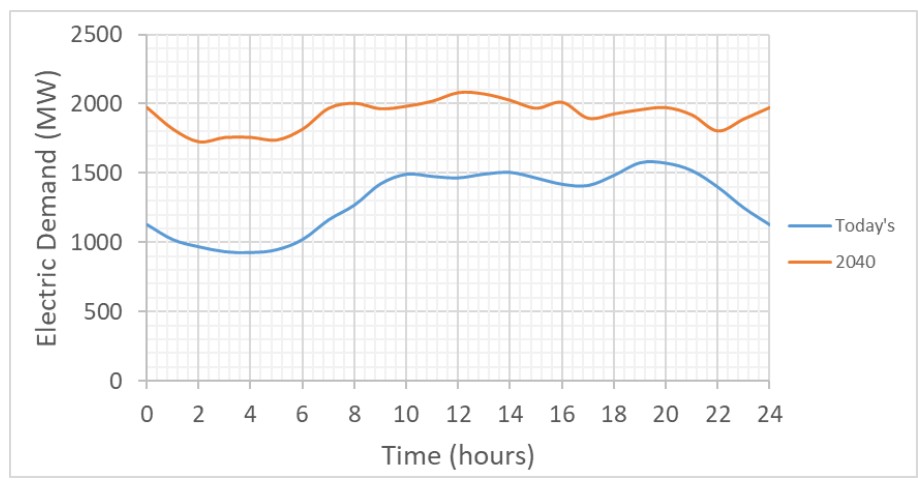

**Figure 4.** Current electricity demand profile and forecast of the Canary Archipelago for 2040.

In relation to the interconnections, these have been dimensioned in such a way that they can allow energy exchanges between islands under any circumstances, as seen in Table 2. To this end, for the islands located at the ends of the grid (Lanzarote and La Palma) the interconnection allows the passage of peak demand plus a 10% safety margin to allow that no extensions should be made (the demand in 2040 will be stabilized, as in fact it is currently, but it will change fundamentally due to the contributions of electric vehicles). For the island of La Gomera, the interconnection currently in the design phase has been assumed, since it has a transport capacity greater than the peak demand foreseen for 2040, plus the 10% safety margin. For the island of Fuerteventura, the interconnection has been sized so that it can transport its demand plus that of Lanzarote. For the two central islands, those with the maximum demand, the interconnection is sized to be

capable of transporting the highest demand peak plus the 10% safety factor; it is not over-sized to a greater degree, given that these two islands are where the bulk of both the generation and storage systems will be located.

**Table 2.** Major characteristics of the interconnection system.

| Interconnection | Power (MW) | Length (km) | Cost (M EUR) [1] | Total Losses (%) [2] |
|---|---|---|---|---|
| Lanzarote–Fuerteventura | 230 * | 16.3 | 270 | 1.07 |
| Tenerife–La Gomera | 50.8 ** | 42 | 124 | 1.17 |
| Tenerife–Gran Canaria | 1050 | 72 | 819 | 1.29 |
| Gran Canaria–Fuerteventura | 400 | 90 | 473 | 1.36 |
| Tenerife–La Palma | 82 | 88 | 183 | 1.35 |

[1] Costs are estimated through Equation (1), and the interconnection between Tenerife La Gomera is projected in 103 M EUR; [2] losses are estimated according to the conservative expression (1 + 0.4 Length/100); * an interconnection project between Lanzarote and Fuerteventura with a power of 121 MVA is being constructed; ** an interconnection project between Tenerife and La Gomera with a power of 50.8 MVA is being planned, higher than the 26.4 of the peak demand of the island plus the 10% of the safety coefficient.

In the estimations of the interconnection lengths between the different islands, the cable distances buried in the land from the substations to the coast have been considered, as well as the length of the underwater route, taking into account the particular underwater orography of each case. As an example, the profile of the interconnection between the islands of Tenerife and La Gomera is shown in Figure 5. This figure shows that the distance between both islands is around 29 km, but the additional distance to bring the cable through the seabed brings the distance up to about 35 km, which together with the distance to reach the substations at both ends leads to the total length of 42 km, as shown in Table 2. Therefore, the same procedure has been followed for the remaining proposed interconnections between the islands.

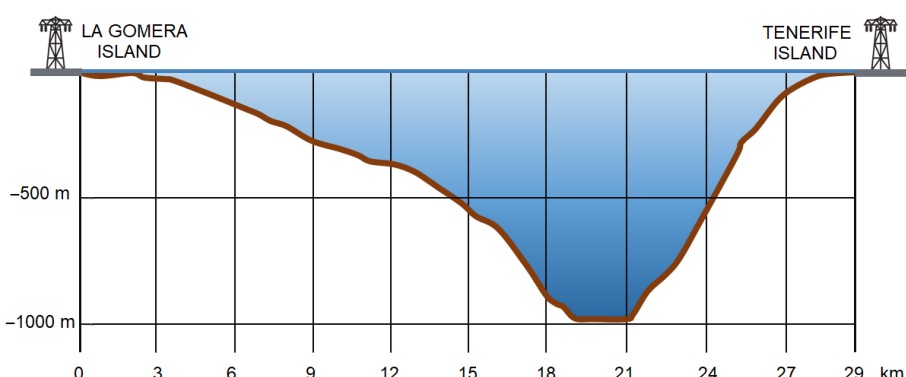

**Figure 5.** Profile of the interconnection between Tenerife and La Gomera islands in 2040.

Qiblawei et al. [30] have proposed a regression model of the cost of interconnections using Bloomberg New Energy Finance (BNEF) data. In the fitting process, they tested several parameters, such as rated power, distance, voltage, and year of commissioning; finally, the authors concluded that the combination of power and distance produces good results, with an $R^2 = 0.85$. The authors estimated that one has a maximum error of 27.8% of the costs predicted by the correlation versus the actual costs based on the cost analysis of 52 HVDC projects. The proposed expression is as follows:

$$C_{SM} = 7.172 \cdot P_{MW}^{0.5989} \cdot L_{km}^{0.1336} \tag{1}$$

where $C_{SM}$ is the cost expressed in million USD (applied exchange ratio 1 USD = 1 EUR), $P_{MW}$ is the maximum power transported through the interconnection in MW, and $L_{km}$ is the interconnection length expressed in km. According to Gils and Simon [34], the installation has a life service of about 40 years and annual fixed operating costs equivalent to 0.6% of the initial investment.

The interconnections between islands are shown in Figure 6, while Table 2 summarizes the main characteristics of the analyzed interconnections between the Canary Islands. Finally, the total losses associated with the interconnection must be considered. These are basically composed of the converter substation losses plus those of the cable itself, which are set at around 1% and 0.4%/100 km, respectively [30]. Gils and Simon [34] assume DC transmission losses of 0.27%/100 km in the marine cables, plus additional losses of 0.7% in the conversion from an AC–DC and DC–AC. Finally, assuming that less than 25% of the total energy generation needs to be transported to other islands (most of the generation is produced, almost equally, between the two islands of greatest importance, namely Gran Canaria and Tenerife), then the total energy losses are around 0.25–0.35% of the generated electricity.

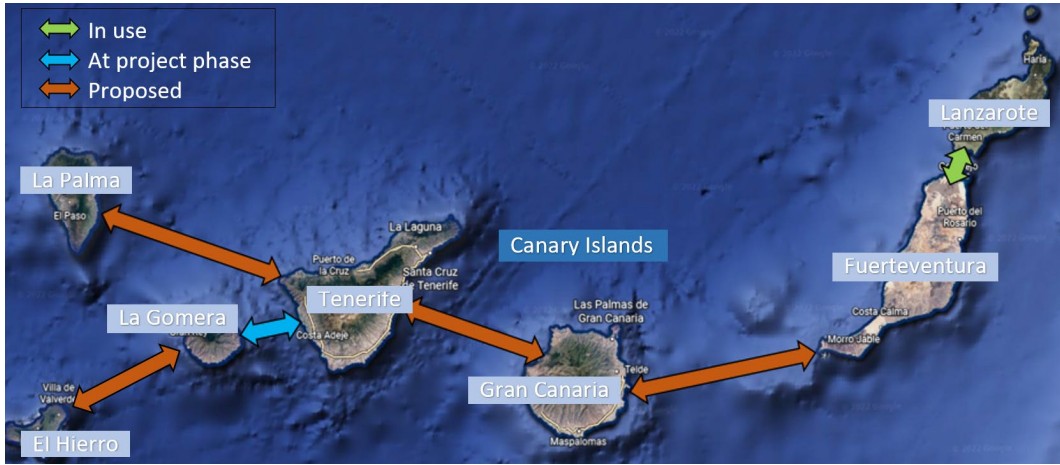

**Figure 6.** Proposed grid interconnection between the different islands of the Canary Archipelago.

The first of the interconnected scenarios consists of considering the aggregate demand, generation, and storage of all the Canary Archipelago islands, so that the software can be used to estimate the optimum mix, highlighting that the costs of the interconnections and their associated losses have been taken into account. In the second interconnected scenario, the different characteristics of the systems used have been analyzed and optimized; for example, among other considerations, the designs of the pumping and biomass plants have been reviewed, and the possibility of increasing the installed capacity of solar PV above the available roof area has been explored. All these aspects will be described and discussed in more detail in the results section.

### 3.3. Simulation Inputs

The generation sources considered are wind and solar photovoltaic, in combination with the unavoidable use of storage (battery systems and reversible pumping storage have been considered) to achieve the optimal design of the generation system. The final results are a compromise solution between cost reduction, excess energy minimization, and affordable land occupation, always covering the demand. In the current cases, the system is conceived to supply 100% of the Canary Archipelago's energy demand. Additionally, the previously mentioned criterion of zero $CO_2$ emissions has also been met, since the nature of all used technologies leads to zero emissions during their operational lifetime. However, if the emissions during their whole life cycle are considered, then there

are emissions, in any case, much lower than those currently existing. Figure 7 presents a scheme of the sources analyzed to cover the energy demand for the considered scenarios.

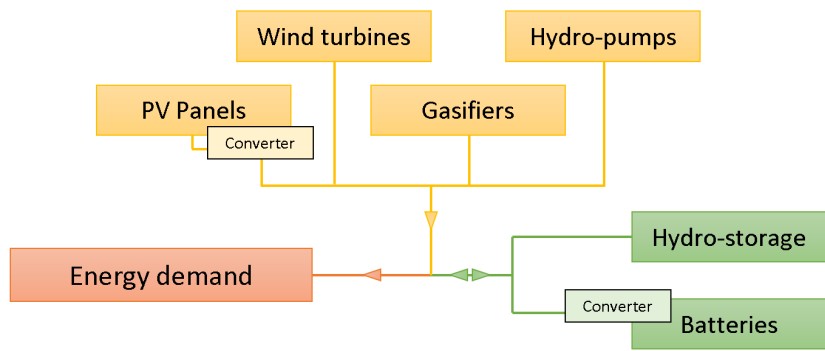

**Figure 7.** Scheme of the energy demand scenarios and the analyzed energy sources.

## 4. Renewable Power Generation System

As already mentioned, decarbonization should be a reality in the European Union by the mid-century. In the non-peninsular territories of Spain, there is a plan to lead the ecological transition and installation of a decarbonized energy system [25]. This plan is the Energy Transition Plan (PTECan), a plan that aims to achieve the economic decarbonization of the Canary Islands by the year 2040. As part to this process, three strategies are also being developed on key aspects of the Canary Islands system, namely self-consumption, electric vehicles, and storage [42–44]. For this reason, all of the Canary Islands are working hard to minimize their high dependence on fossil fuels. However, they have an enormous advantage given the abundant natural resources of the archipelago, particularly the wind and the sun. Furthermore, the Canary Islands benefit from the presence of several good sites for reversible pumped storage power plants due to the volcanic origin of the islands.

### 4.1. The Photovoltaic System

Given the characteristics of the existing radiation in the Canary Islands, the solar resource can be an essential source of energy generation in a renewable system. In fact, the archipelago has the highest insolation in Spain. This solar resource can be estimated through NASA's POWER Data Access Viewer [45]. According to its documentation, the hourly solar data are based on satellite observations together with solar surface irradiance information from NASA's Global Energy and Water Exchange Project (GEWEX)/Surface Radiation Budget (SRB) Release 3, and NASA's CERES fast longwave and shortwave radiative project (FLASHFlux).

Initially, it was planned to sample the last 10 years to obtain average hourly values. However, this would produce an attenuation of the resource variability, so the year 2019 was finally used. This choice is motivated by the high climatic stability of the islands, having a subtropical climate, so that temperature changes are very low, since water is a great thermal regulator. The climate of the Canary Islands is arid, with low rainfall, which makes the annual irradiation quite similar year after year. This information has been obtained for each island, although it is very similar for all of them. As an example, Figure 8 shows the monthly solar energy resource of the Gran Canaria Island. The measure of the clarity of the atmosphere is the clearness index; this variable is defined as the fraction of solar radiation that is transmitted across the atmosphere to impinge on the Earth's surface [35]. This energy represents a potential global horizontal irradiance of 1826 ESH/year (equivalent sun hours), a value that can be increased up to 2442 ESH/year by using solar trackers. These data are the ones used in current research, and it is assumed that this irradiation is maintained until the year 2040. Table 3 provides additional information on the inputs used in the solar PV facility.

Finally, a maximum value of solar PV energy to be installed has also been considered. The capacity to be installed in each of the archipelago's islands is obtained from the optimal estimates of the self-consumption analysis of the last Canary Islands report [44]. For example, according to this report, the photovoltaic solar generation potential on the Gran Canaria Island is 3700 MW, considering a maximum occupancy of 70% of the available roof area on the island (~53 km²). Specifically, the Deloitte Monitor report [25] raises the need to maximize self-consumption along with the installation of solar plants. Then, this capacity could be increased either through the installation of solar farms (not a very recommendable option in some places, since the Canary Archipelago is very touristic and has large areas of national parks and protected areas, so there could be problems of land occupation) or by using part of the available surface in the dams of the reversible pumping installations (floating photovoltaic plant).

**Table 3.** Inputs used for the PV system [46].

| | |
|---|---|
| Lifetime (years) | 25 |
| Derating factor (%) | 90 |
| Tracking system | No tracking |
| Used panel | Vertex 550+ |
| Temperature coefficient of power (%/°C) | −0.38 |
| Peak power (W) | 550 |
| Nominal operating cell temperature (°C) | 45 |
| Efficiency of the panel at standard conditions (%) | 21.1 |
| Cost (EUR/kW) | 1300 |
| O&M cost (per 1 MW peak power) (EUR/year) | 3500 |

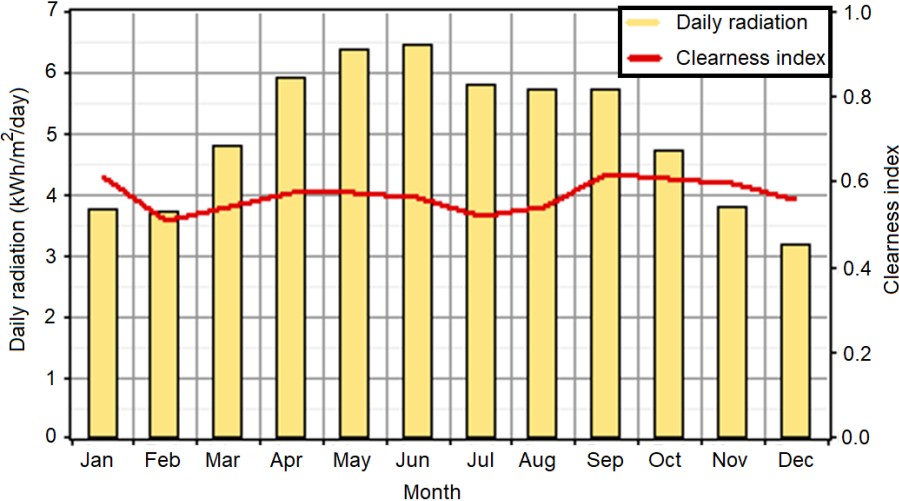

**Figure 8.** Monthly solar energy resource in Grand Canary [47].

*4.2. The Wind System*

The wind in the Canary Archipelago is the other major available resource. The magnitude of the wind resource can be assessed using the global wind POWER Data Access Viewer, developed by NASA [45]. The meteorological information provided by this database is based on the Goddard's Global Modeling and Assimilation Office (GMAO) modern-era retrospective analysis for research and applications (MERRA-2) assimilation model products and on the GMAO forward processing instrument teams (FP-IT) GEOS 5.12.4 near-real-time products. Following the reasoning used for the solar resource, wind data have been consulted for the last 10 years, but averaging has also been discarded. The hourly data for the year 2019 have been finally selected as representative. In fact, as

already mentioned for the solar resource, the wind resource is very abundant, with very good characteristics in terms of wind frequency and intensity, which remain at very similar values year after year. This information is provided to the code to quantify the available wind resource, since it is assumed that these wind data will be maintained until the year 2040.

An analysis of the most suitable sites for the different islands has been carried out, and a view of the average wind speed of the archipelago is shown in Figure 9. As shown, wind generators can be installed in many suitable locations, as both onshore and offshore technologies. As commented earlier, in this study, it has been considered that the best option was the use of offshore technology from the land occupation side. This technology is much more expensive but, given its higher energy production (more stable and higher average wind speeds at marine sites) and from the aforementioned criterion of land occupation, it has been considered the best option. A summary of the datasheet for the selected wind generator is shown in Table 4.

**Table 4.** Datasheet of the wind turbine [48].

| Wind Generator | Enercon E-126 |
| --- | --- |
| Rated power (MW) | 7.58 |
| Rotor diameter (m) | 127 |
| Height to the axe (m) | 135 m |
| Total height (m) | 197 m |
| Lifetime (years) | 25 |
| Cost of the system (M EUR/turbine) | 17.9 |
| M EUR/MW | 2.39 |
| O&M cost (M EUR/year) | 3.5 |

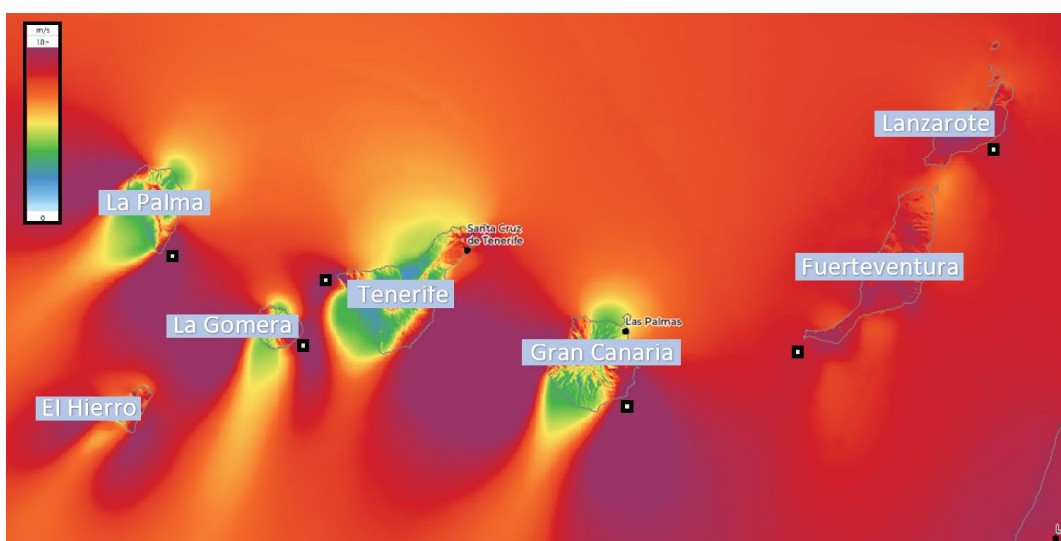

**Figure 9.** The offshore wind resource and optimum wind farm sites in the Canary Islands [49].

### 4.3. The Storage System

The use of storage systems provides greater security and flexibility in the energy supply for any generation system. Furthermore, they become essential in isolated regions with stand-alone grids, such as islands, when aiming to manage the unavoidable electric energy surpluses when a full renewable system is implemented [30,50]. Various energy storage groups of technologies have been developed to absorb these electricity surpluses, e.g., the conversion of electrical energy into mechanical, thermal, gravitational, electrochemical, and chemical energy. However, there are currently only two storage

technologies which can provide the storage capacity required for the systems discussed in this study, namely reversible pumped storage plants and mega-batteries.

4.3.1. Reversible Pumped Storage

This technology is quite mature, and there are several plants all over the world. More specifically, there is a reversible pumped storage hydroelectric power plant already planned for one of the Canary Islands, namely the Chira-Soria project [27] on Gran Canaria, which is expected to be operational by 2026–2027. The Chira-Soria pumping station will have a total storage capacity of about 3.2–3.6 GWh, with an output power of 200 MW, equivalent to 16 h at full power if the upper dam is completely full. The budget to manage the Chira-Soria project is approximately 400 million EUR.

The substantial initial capital investment of reversible pumped storage plants can be partially mitigated by their long service life of up to 75 years or possibly longer. Several studies have shown that the life of a hydroelectric power plant is, at least, about 50 years [51], or that they can even reach lifetimes of 100 years [52]. Another advantage of these technologies is the reduced maintenance cost, around 2% of the initial investment per year [51]. For these reasons, reverse-pumped storage is the most widely used technology for electricity storage (approximately 95% of all storage facilities in the word). The energy efficiency of the complete round-trip cycle (turbine and pumping) ranges from 70% to 85% [53,54].

In relation to the Canary Islands, given the orography of many of them, they have a certain number of locations suitable for pumping [43]; Table 5 shows a list of the installable power and total storage capacity broken down by islands. It should be recalled that, on the island of Gran Canaria, the first reversible pumping plant project is being carried out, namely the Chira-Soria project, with planned values of 200 MW for turbinating power and 3.2–3.6 GWh of stored energy.

**Table 5.** Summary of the estimated reverse pumping storage capacity in the Canary Islands [43].

| Island | Power (MW) * | Energy (MWh) ** |
|---|---|---|
| **Tenerife** | 234 | 4600 |
| **Gran Canaria** | 607 | 9800 |
| **La Palma** | 104 | 1450 |
| **La Gomera** | 64.5 | 786 |

\* Total turbinating power (pumping power is approximately 10–20% higher); ** Total energy that can be fed into the grid.

As for the three remaining islands, the two eastern islands of the Canary Archipelago are very arid, which would make the use of pumping technologies difficult. In Fuerteventura, there are only three reservoirs of considerable size to be used for energy purposes. These reservoirs are located at great distances from each other and with relatively little difference in elevation, so it would not be an option to interconnect them. Something similar occurs in Lanzarote, where there is only one dam. Therefore, the only alternative in both cases would be to build a second reservoir nearby, which would make the installation more difficult and expensive.

Quite a similar situation happens on the island of El Hierro, where there are only three dams. The existing distances between these reservoirs make it unfeasible to propose alternatives analogous to those analyzed for the rest of the islands. In any case, solutions similar to the one proposed for the Gorona del Viento hydro-wind power plant could be implemented, although the island now has the facilities to be self-sufficient. Therefore, this study has not considered its interconnection with the rest of the islands, given its energetic self-sufficiency, remoteness, and low demand (0.5% of the total of the islands), which does not make it either profitable or necessary.

For the simulation of reverse pumping stations that cannot be simulated directly in the HOMER code, an alternative had to be developed, and this alternative consists of the design of a hydrogen storage system (hydrogen tank, electrolyzer, and generator), as suggested by Berna et al. [41]. In the designed system, the maximum electricity consumption of the electrolyzer is, by analogy, the pump drive power. The capacity of the hydrogen reservoir corresponds to the total capacity of the upper reservoir (potential energy stored from the height difference between reservoirs when the upper reservoir is full that will be the same as the amount of hydrogen stored to contain that energy). Meanwhile, the generator coupled to the hydrogen tank (fuel cell or hydrogen combustion) simulates the maximum turbine power.

### 4.3.2. Mega-Batteries

The storage of electrical energy without undergoing intermediate transformations is directly associated with electric batteries. The use of this technology is being favored fundamentally by the need for an electrical energy storage capacity motivated by the increasing use of electric vehicles, as well as by the recent development of high-capacity batteries associated with the great storage needs of the inevitable excess generation caused by the growing penetration of renewable energies. As a result, electric battery technology has undergone major development, and costs are being significantly reduced. However, it is still necessary to increase their useful life and to promote the reuse and recycling of their components. However, the current trend of continuous improvement will continue in the coming years.

Consequently, an additional storage capacity is necessary in case not all requirements can be covered by the total reverse pumping capacity. As discussed above, the technology that is currently at a higher state of maturity and which seems the only alternative today is the use of mega-batteries. The operation would be the same as that of pumping stations, absorbing excess energy, and then returning it when necessary. Table 6 summarizes the specifications of the mega-battery system used, a Gildemeister 250 kW-8 h Cellcube.

**Table 6.** Standard system specifications of the selected battery system [55].

| Battery | Gildemeister |
|---|---|
| Maximum AC power (kW) | 250 |
| Energy available (MWh) | 2.480 |
| Round-trip system Efficiency | 70% |
| Cost of the module (EUR) | 460,000 |
| O&M cost (EUR/year) | 5300 |
| Lifetime (years) | 25 |

### 4.4. The Biomass System

The main sources of biomass for energy production are wood and wood-based products. Some other types of organic material commonly used as fuel in biomass power plants are agricultural vegetable and animal by-products, organic matter from urban waste, human and animal manure, etc.

A backup system that can be used in fully renewable systems, such as the one proposed in this work, and which is also renewable, can be the use of biomass to produce electricity. A biomass power plant is an industrial facility that converts organic matter into usable energy. Biomass power plants can use several processes to convert the energy contained in biomass. The first is direct combustion (burning the biomass to produce heat), and other systems can be microbial conversion (anaerobic bacteria digest and ferment the biomass to produce gas and alcohol), pyrolysis (heat input in the absence of oxygen to produce gaseous fuels), gasification (heat input with controlled amounts of oxygen and steam to produce gas and liquid fuels), and chemical conversion (organic fats and oils are

transformed into methyl esters for the production of biodiesel fuel). However, only two ways are possible commercially, and these are direct-fired power generation and gasification power generation. Of these, combustion is the most widely used process to produce heat and electricity from biomass [56], but this process is more suitable when continuous production is desired; thus, in the current case study this system will be used as a backup system, as gasification is more suitable due to its versatility. In addition, combustion processes generally have higher emissions than gasification, particularly of greenhouse gases (about 30%) [56].

In the gasification process, the thermochemical conversion of solid biomass into a gaseous fuel is carried out by the reaction between the solid biomass and a gasification agent (air, oxygen, or steam water) at high temperatures (700–900 °C) [56]. There are many different gasifier designs, but the objective of all of them is to obtain a gaseous fuel, usually called synthesis or production gas, which is mainly composed of carbon monoxide (CO), carbon dioxide ($CO_2$), hydrogen ($H_2$), and methane ($CH_4$) [57]. This synthesis gas (syngas) has various uses, such as in applications requiring heat and/or electricity generation [58]. It can be employed for heat applications where high and low temperature industrial heat is required, or for electricity generation. The electrical energy may be produced by internal combustion engines, such as the Otto cycle or diesel cycle, or in gas turbines using the Brayton cycle. Furthermore, this gas can be utilized as fuel in molten carbonate fuel cells (MCFCs) and solid oxide fuel cells (SOFCs), both of which have higher conversion efficiencies and are able to operate in combined heat and power (CHP) mode using a biomass gasification combined cycle (BGCC) system. Other applications of synthesis gas include the generation of several biofuels for the transportation sector, such as methanol, hydrogen, dimethyl ether (DME), syngas (SNG), and Fischer–Tropsch (FT) diesel.

Finally, the power generation performance of these gasification systems with combined heat and power (CHP) can reach efficiencies in conditions of electrical energy conversion that could be even higher than 35% [58]. However, in existing practical applications, such as the Vaasa gasification plant in Finland, with an output of 140 MWe and a cost of 40 M EUR (as of the year 2013), the electrical efficiency is 25% (the plant is combined with the Vaskiluoto 2 coal plant, with an electrical output of 230 MW, and the combined electrical efficiency of the two plants is 40%) [57]. Although in Chowdhury's thesis [59], which has a detailed analysis of different biomass gasification systems, the author says that gasification plus gas engine could lead to electrical efficiencies between 22–35% with costs around 6.5–14 EUR/Mwe, while integrated gasification combined cycle technologies (IGCC) could reach efficiencies of 40–50% with electricity costs of around 10.5–13.5 EUR/MWe, this second technology is applicable for large plants (30–100 MWe), even though both technologies are not yet at an advanced commercial stage.

In the case analyzed in this study, biomass will be used primarily as a backup energy source, so that, together with storage systems, it will be responsible for providing the necessary reliability of supply that wind and solar PV cannot provide due to their intrinsic variability. For this reason, among the different technologies that use biomass as fuel, the design of a conventional biomass-fired thermal power plant has not been selected. As previously mentioned, these facilities are the most widely used, but they are more suitable for continuous operation, but since their use will be mainly for backup purposes, the gasification technology will be more appropriate. In this sense, the gasifier produces the gas, which is stored, and when needed is ready to be used, for example, in a gas turbine.

Focusing on the biomass supply potential of the Canary Islands, as explained by [60], these islands store a significant amount of energy in the form of biomass. It would be possible to design small power plants with fuels from agricultural activities. The problem arises when supply to larger consumption centers has to be guaranteed, that is when confronted with the installation of larger power plants that have a greater efficiency, and in terms of demanding the availability of resources with regularity, quality, and at an acceptable cost. Additionally, the majority of the Canary Islands have a very rugged orography, which makes the use of biomass more difficult and significantly more expensive.

Although, the theoretical potential of forest biomass in the Canary Islands, considering only that produced by pine and laurel forests, can be estimated at 5500 toe/year, the biomass coming from agricultural residues is added to this potential, meaning that approximately 15,000 toe/year could be reached [60]. In other words, there is a total of around $6.0 \cdot 10^4$ tons per year of biomass if only the resources existing in the forests are considered, and a total of around $1.63 \cdot 10^5$ tons per year if agricultural residues are added.

## 5. Results and Discussion

The following paragraphs summarize the major results of the three scenarios. For each one of them, the economic and technical aspects will be considered, analyzed, and discussed.

With the different existing resources, specifically wind, sun, and biomass, added to the conditioning factors previously mentioned (maximum available capacities of reversible storage, solar PV, and biomass), the generation plus storage system necessary to cover the demand in its entirety has been optimized. Given the large storage capacity required to manage a fully renewable system of the scale addressed, extra storage capacity has been required (limited locations exist on the different islands to install the necessary infrastructure for the reversible pumping plants), which has been met by the implementation of mega-battery storage systems. In other words, the power to be installed for wind, solar PV, and biomass has been determined, together with the power and storage capacities of the reversible pumping and batteries to make the system self-sufficient. These calculations have been made for the three scenarios considered, namely the one in which each island must be able to cover its own demand (six stand-alone systems) and the two scenarios in which the islands are interconnected.

### 5.1. Technical Characteristics of the Systems

#### 5.1.1. Non-Interconnected System

Tables 7 and 8 summarize the power and energy covered by each energy source with the proposed mix for each island of the Canary Archipelago. As shown in the tables, only renewable sources (wind, solar PV, and biomass) have been utilized as generation sources. The highest installed power is, by far, solar PV, at approximately 85% of the 12.6 GW installed. The two smallest islands have slightly different power distributions, as La Palma has a proportionally lower solar PV (~75%) and higher wind and biomass (~14 and 12%, respectively), while La Gomera has the opposite values, namely a proportionally higher solar PV (~93%) and lower wind and biomass (~7 and 0.1%, respectively).

**Table 7.** Summary of the installed generation power and storage power and capacity.

| | Tenerife | Gran Canaria | Lanzarote | Fuerteventura | La Palma | La Gomera | Total |
|---|---|---|---|---|---|---|---|
| Renewable sources | | | | | | | |
| Solar PV (MW) | 5000 | 3700 | 906 | 750 | 300 | 100 | 10,756 |
| Wind (MW) | 675 | 480 | 142.5 | 112.5 | 60 | 7.5 | 1478 |
| Biomass (MW) | 200 | 100 | 10 | 6 | 50 | 10 | 366 |
| Total power (MW) | 5875 | 4280 | 1058.5 | 868.5 | 410 | 117.5 | 12,600 |
| Storage systems | | | | | | | |
| Hydro pump (MW) | 234 | 607 | - | - | 104 | 64.5 | 1010 |
| Battery system power (MW) | 1750 | 1000 | 500 | 375 | 38 | - | 3663 |
| Total power (MW) | 1984 | 1607 | 500 | 375 | 142 | 64.5 | 4673 |
| Pumped storage capacity (GWh) | 4600 | 9800 | - | - | 1450 | 786 | 16,636 |
| Battery system capacity (GWh) | 17,336 | 9906 | 4953 | 3715 | 371 | - | 36,282 |
| Total energy (GWh) | 21,936 | 19,706 | 4953 | 1715 | 1821 | 786 | 52,918 |

**Table 8.** Energy production per component and island.

| | Tenerife | Gran Canaria | Lanzarote | Fuerteventura | La Palma | La Gomera | Total |
|---|---|---|---|---|---|---|---|
| Renewable sources | | | | | | | |
| Solar PV (GWh) | 11,817 | 8562 | 2025 | 1672 | 716 | 239 | 25,030 |
| Wind (GWh) | 2023 | 3373 | 649 | 507 | 193 | 22 | 6767 |
| Biomass (GWh) | 144 | 1 | 15 | 8 | 15 | 0.5 | 183 |
| Total energy (GWh) | 16,403 | 13,732 | 3276 | 2618 | 1150 | 350 | 31,980 |
| Storage systems | | | | | | | |
| Hydro pump (GWh) [1] | 646 | 1370 | - | - | 203 | 88 | 2307 |
| Battery (GWh) [1] | 1774 | 427 | 587 | 431 | 23 | - | 3242 |
| Total energy (GWh) [1] | 2420 | 1797 | 587 | 431 | 226 | 88 | 5549 |
| System excesses | | | | | | | |
| Surpluses (%) | 36.2 | 34 | 28.5 | 32.0 | 22.4 | 17.9 | 33.8 |

[1] Energy re-fed to the grid by the storage systems; this energy comes from the renewable generation.

As for the storage system, the total installed power is approximately 4.6 GW, with an energy storage capacity of around 51 GWh. As Lanzarote and Fuerteventura do not have adequate sites for reverse pumping stations, then these deficits have to be compensated by the installation of mega-batteries. Tenerife has a reduced availability of locations for the installation of reversible pumping plants, so it must also resort to the installation of a large number of mega-batteries. The islands of Gran Canaria, La Palma, and La Gomera have higher power percentage and, thus, require lower capacities of mega batteries; in fact, La Gomera does not need to resort to this technology, and La Palma needs a reduced power storage system. The energy amount that passes through the storage systems and re-feeds to the grid, as shown in Table 8, is significant. For instance, Tenerife's system is capable of recovering approximately 15% of the total generated energy; the same happens with Gran Canaria, while Lanzarote, Fuerteventura, and La Palma recover almost 20%, and La Gomera recovers around 25%. All these figures are after considering the storage system losses.

As mentioned in a previous section, biomass is mainly used as a support system, so efforts have been made to maximize its use, always considering the limited resources existing on the islands. In this case, as can be seen in Table 8, the production of energy through biomass is very limited, and it is only appreciable in Tenerife (around 1% of the total energy produced), and to a lesser extent in La Palma and Lanzarote; it is practically residual for the rest of the islands. However, this production, although small, is important because it is used when no other energy is available, acting as a backup in case of any contingency. Therefore, it is important to maintain this generation source, on the one hand because it is a backup source and, on the other, because it takes advantage of the islands' biomass resources, which if they were not collected to be used as biomass would not be used and would simply be burned.

One of the most important points to take into account is that, despite the large capacity of the storage systems, there are substantial electricity surpluses (a weighted average of 33.8% of the 32 TWh per year produced) in all islands except the two smallest ones, which have around 20%, but they represent a very small part of the total amount. The major cause of these important surpluses is motivated by the high percentage weight of solar generation in the proposed mix, which, as detailed in the economic analysis, is due to its lower installation, operating, and maintenance costs. Solar PV generation represents more than 85% of the installed power while generation represents around 78%, wind installed power is slightly below 12% and represents more than 21% of the generation, while biomass is almost 3% of the installed power and generates a residual percentage, around 0.5% (although its importance as backup system must be remembered). The storage systems can return over 17% of the generated electricity to the grid. It must be highlighted that these percentages of surpluses are quite high but, considering that this generation

system is a stand-alone system and based on renewable sources, these figures could be considered as normal.

To have a general idea of the behavior of the generation–storage system, a complete annual representation of each generation and storage source is displayed in Figures 10–12. These figures correspond to data from the island of Tenerife, but the rest of the islands with reversible pumping and mega-battery installations show similar behavior (Gran Canaria and La Palma), having some differences on the other islands. For example, Lanzarote and Fuerteventura, since they do not have reversible pumping, cover the lack of energy, mainly at night, by means of the mega-battery system. On the other hand, La Gomera can cover the totality of hours with a generation defect by means of reversible pumping. Returning to the aspects related to the energy generation sources, shown in Figure 10, the solar resource obviously has a greater contribution in the summer months (Figure 10a), both in power and hours (days 120 to 240 approximately, coinciding with the months of May to August), being one of the main renewable resources of the Canary Islands given its privileged location. In relation to the wind resource, Figure 10b, it is observed that the major contributions also occur in the summer months (days 170–240, which approximately coincides with the months of July and August), and that the minimums occur between days 270 and 300 of the year (month of October).

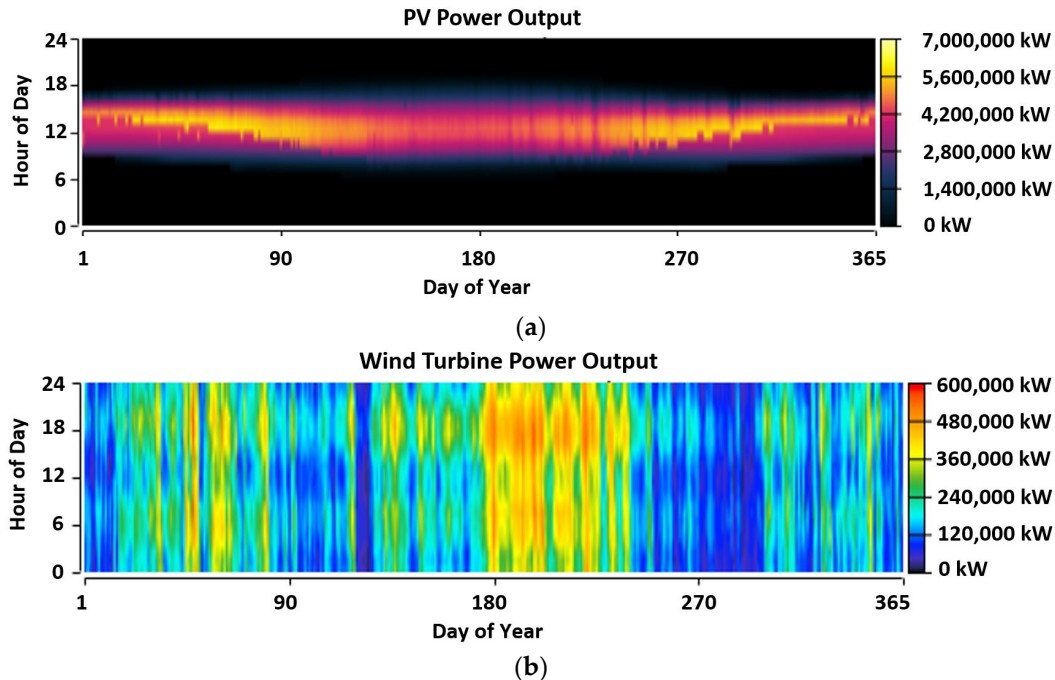

**Figure 10.** Yearly performance of the energy sources. (**a**) Solar PV; (**b**) wind.

In relation to the use of a reversible pumping system, Figure 11 displays the pumping and turbinating cycles throughout the year. In the pumping refill cycles (Figure 11a), it is observed that during the summer months there are a greater number of hours, with practically all of them pumping at maximum power, so that a higher level is reached in the upper tanks. As for the grid re-feed of the reversible pumping, it should be noted that, as can be seen in Figure 11b, it is only available during the first hours of the night for the winter months, in fact, until approximately midnight or even earlier. As spring progresses, this contribution gradually increases over time, until it reaches its maximum in the summer months, and then decreases again to the previously commented values as winter approaches. Figure 12 is displays the yearly performance of the charge–discharge cycles of the mega-battery system. In the case of the charging cycles of the batteries (Figure 12a), it is clearly observed that in the summer months the charging time is much lower; this is because they are used only when the pumping capacity has been exhausted. Thus,

they are close to full charge, while in winter they have a higher degree of demand, which causes them to have a greater discharge, with the consequent need to recharge at times of production excess. In relation to the grid re-feed of the mega-battery system, Figure 12b, it must be said that it controls the demand coverage during the intervals where the pumping is not able to, so that in winter it covers from midnight to the early hours of the morning, while in summer its contribution is reduced to the last hours of the night and first hours of the morning.

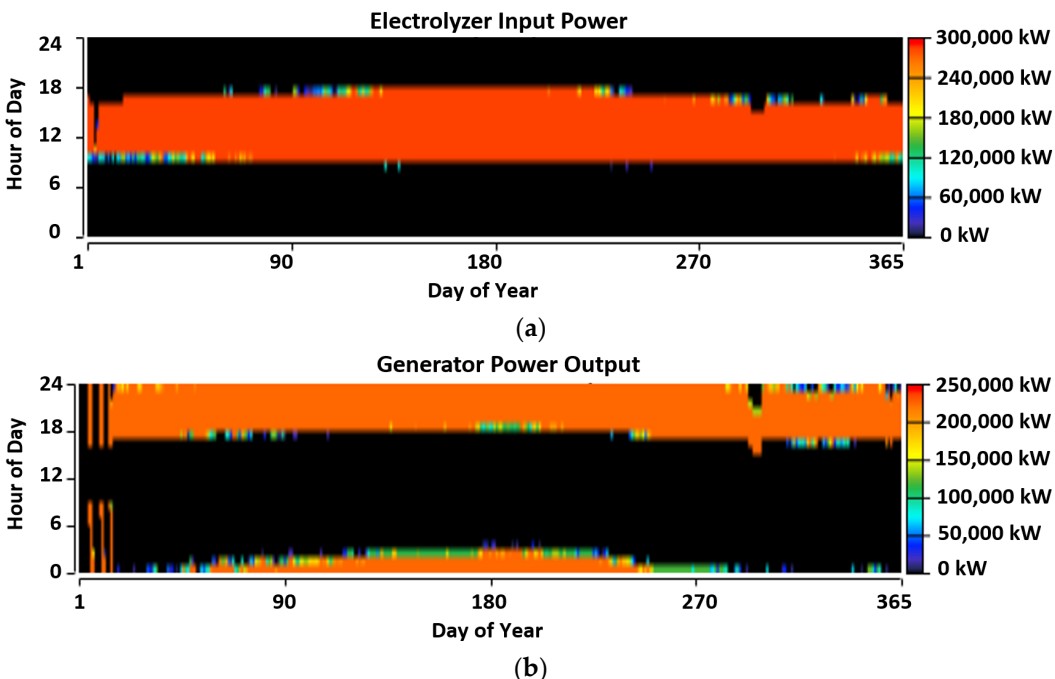

**Figure 11.** Yearly performance of the reverse pumping system. (**a**) Pumping; (**b**) turbination.

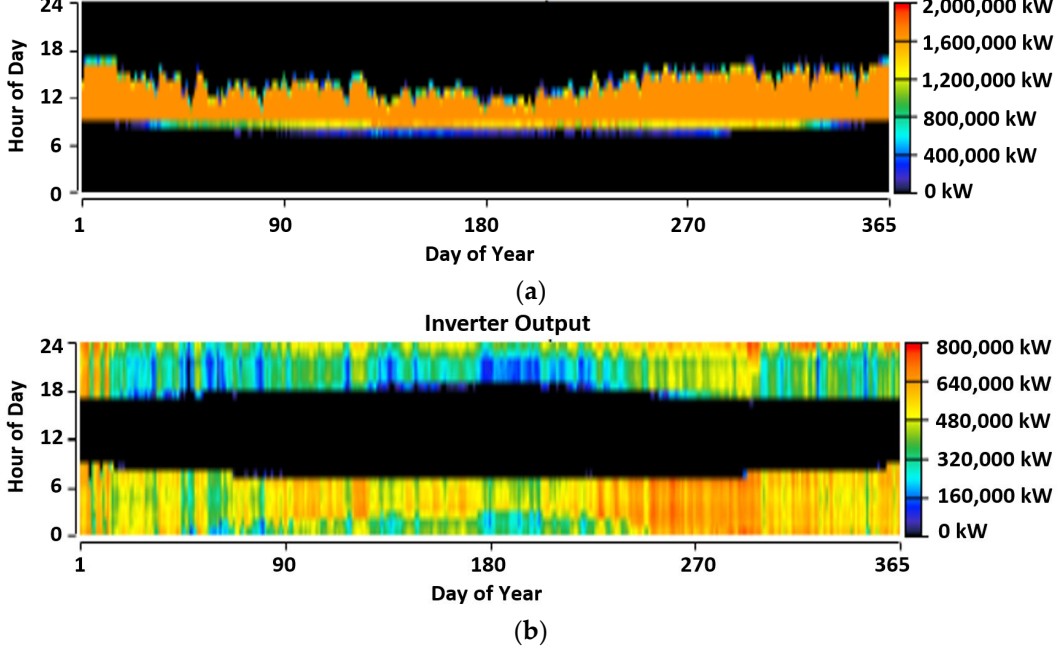

**Figure 12.** Yearly performance of the mega-battery system. (**a**) Charge; (**b**) discharge.

In relation to the storage systems, Figures 13 and 14 show the monthly data diagrams. La Gomera is the only island where the existing reversible pumped storage capacity has

been sufficient to manage the fully renewable system (as shown in Figure 13d). In fact, the system is at its maximum level in almost all months (i.e., the system has been oversized to cover the energy shortages in January). Thus, this oversizing will be an advantage in the interconnected scenarios, as will be seen in the corresponding sections (Sections 5.1.2 and 5.1.3), since this high pumping storage capacity will be used in the interconnected systems to cover the shortages of other islands. Lanzarote and Fuerteventura do not have suitable locations for the installation of pumping plants, so it has been necessary to establish a high storage capacity by means of mega-batteries in proportion to their demand. As displayed in the Figure 14c,d, October is the most critical month and the one that has required the storage system to be employed. In La Palma, it is observed that the existing reversible pumping capacity on the island is not sufficient (Figure 13c), except in the summer months of June, July, and August. As such, it is clear that extra storage capacity is needed, to the point that almost in all the winter months (October to March) even the batteries are almost completely discharged at some points (Figure 14e). As for the two islands with the highest demand, both require the combination of pumping storage with the use of mega-batteries, being more restrictive in the case of Tenerife, where there is less pumped storage capacity. In the case of Gran Canaria, the pumping system would be sufficient to manage excess generation only in summer, while for practically all the remaining months, extra storage capacity is required. As can be seen in Figure 14b, it is the month of October that determines the capacity of the necessary batteries.

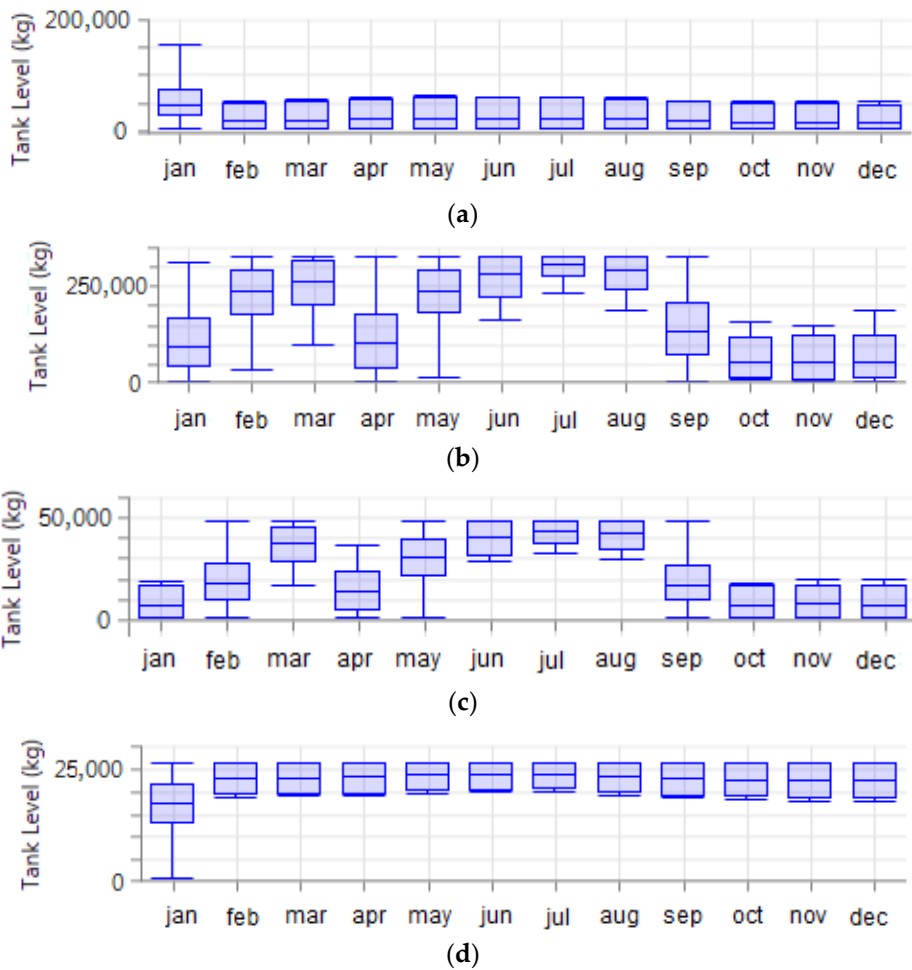

**Figure 13.** Monthly data of the reverse pumping storage system. (**a**) Tenerife; (**b**) Gran Canaria; (**c**) La Palma; (**d**) La Gomera.

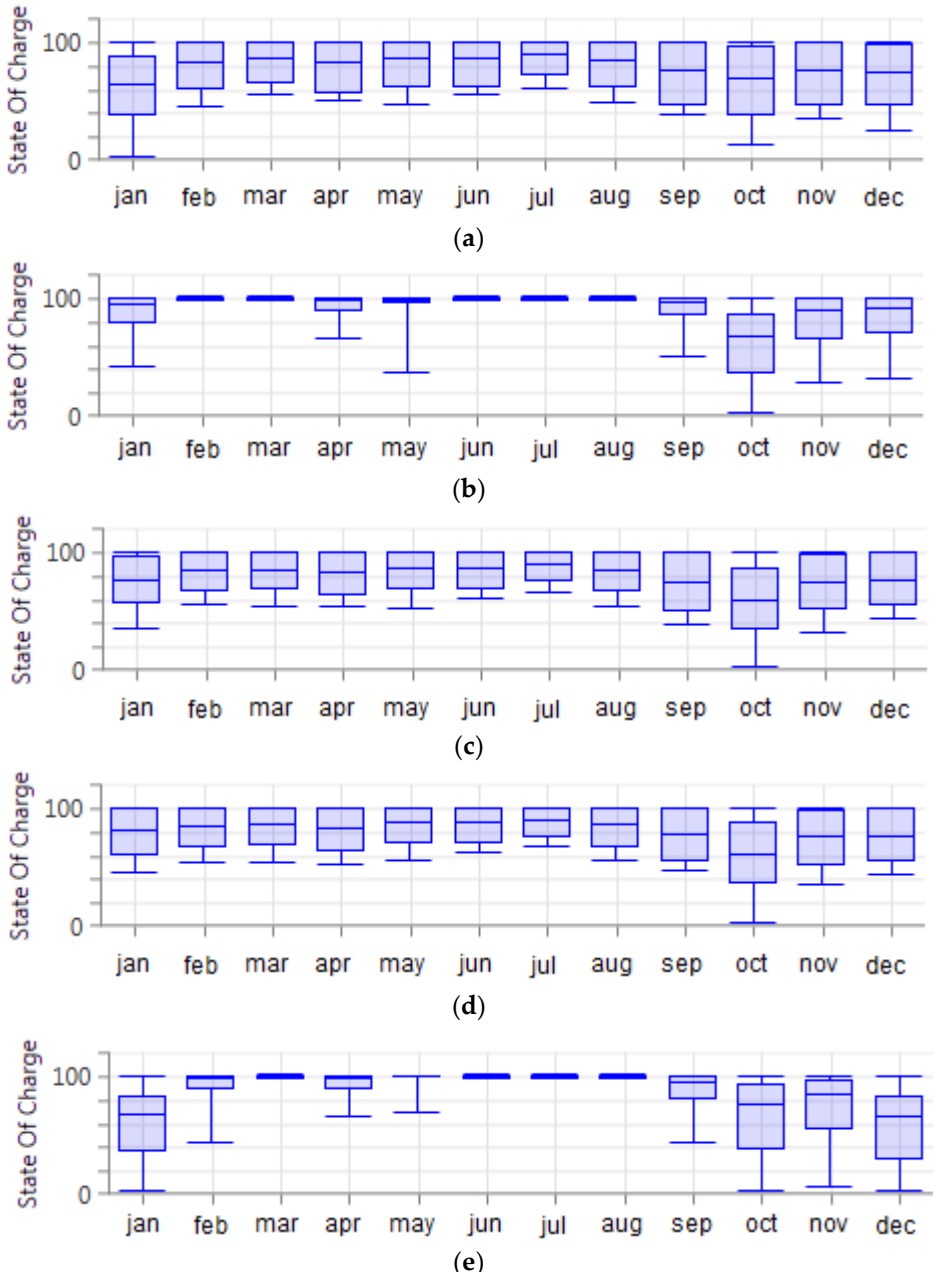

**Figure 14.** Monthly data of the mega-batteries storage system. (**a**) Tenerife; (**b**) Gran Canaria; (**c**) Lanzarote; (**d**) Fuerteventura; (**e**) La Palma.

Additionally, it has been decided to show an hourly representation of typical summer and winter days (Figures 15 and 16), which represent, respectively, the least and most adverse situations for the system. It must be pointed out that, given that most of the generation contribution comes from solar PV energy, obviously the lower radiation in the winter months determines the necessary generation and storage capacities. It should be noted that for the island of La Gomera, even in winter, the existing high pumped storage capacity can recover the excess of energy produced (Figure 16f). The installation of extra storage in the form of mega-batteries is not required here, highlighting that this is the only island where this situation occurs. The opposite is the case on the islands of Lanzarote and Fuerteventura, where there is no reversible pumping storage, meaning that the mega-batteries must absorb the excess energy generated and then provide it during the intervals of power failure throughout the year (Figures 15c,d and 16c,d). In the three remaining islands, both storage technologies must be used during the winter (Figure 16a,d,e), while in

summer, neither La Palma nor Gran Canaria resort to the use of batteries (Figure 15b,f). Tenerife is has to resort to the use of mega-battery storage both in summer and winter (Figures 15a and 16a), as do Lanzarote and Fuerteventura, which do not have a pumping system (Figures 15c,d and 16c,d).

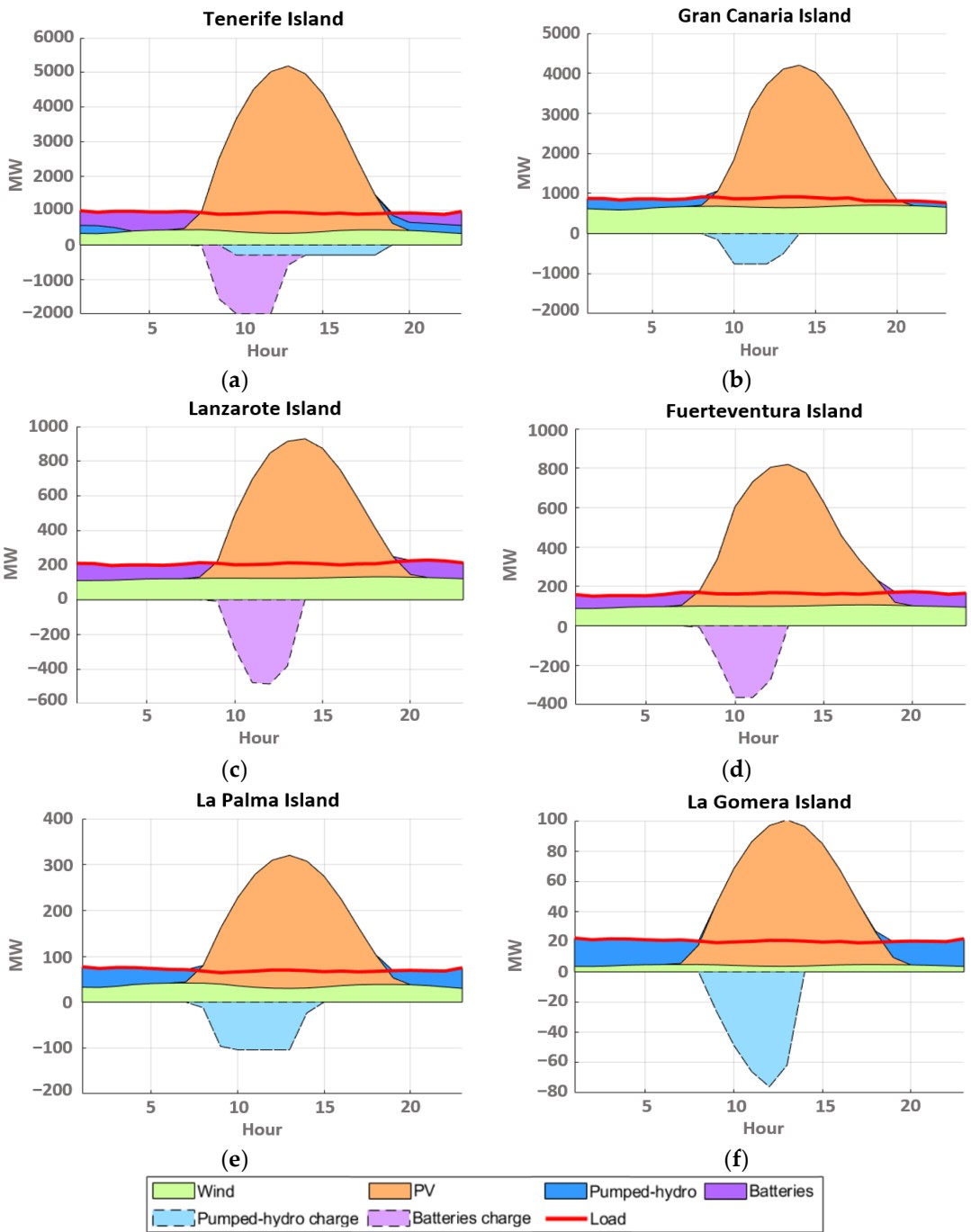

**Figure 15.** Hourly performance of the generation and storage systems on a typical summer day. (**a**) Tenerife; (**b**) Gran Canaria; (**c**) Lanzarote; (**d**) Fuerteventura; (**e**) La Palma; (**f**) La Gomera.

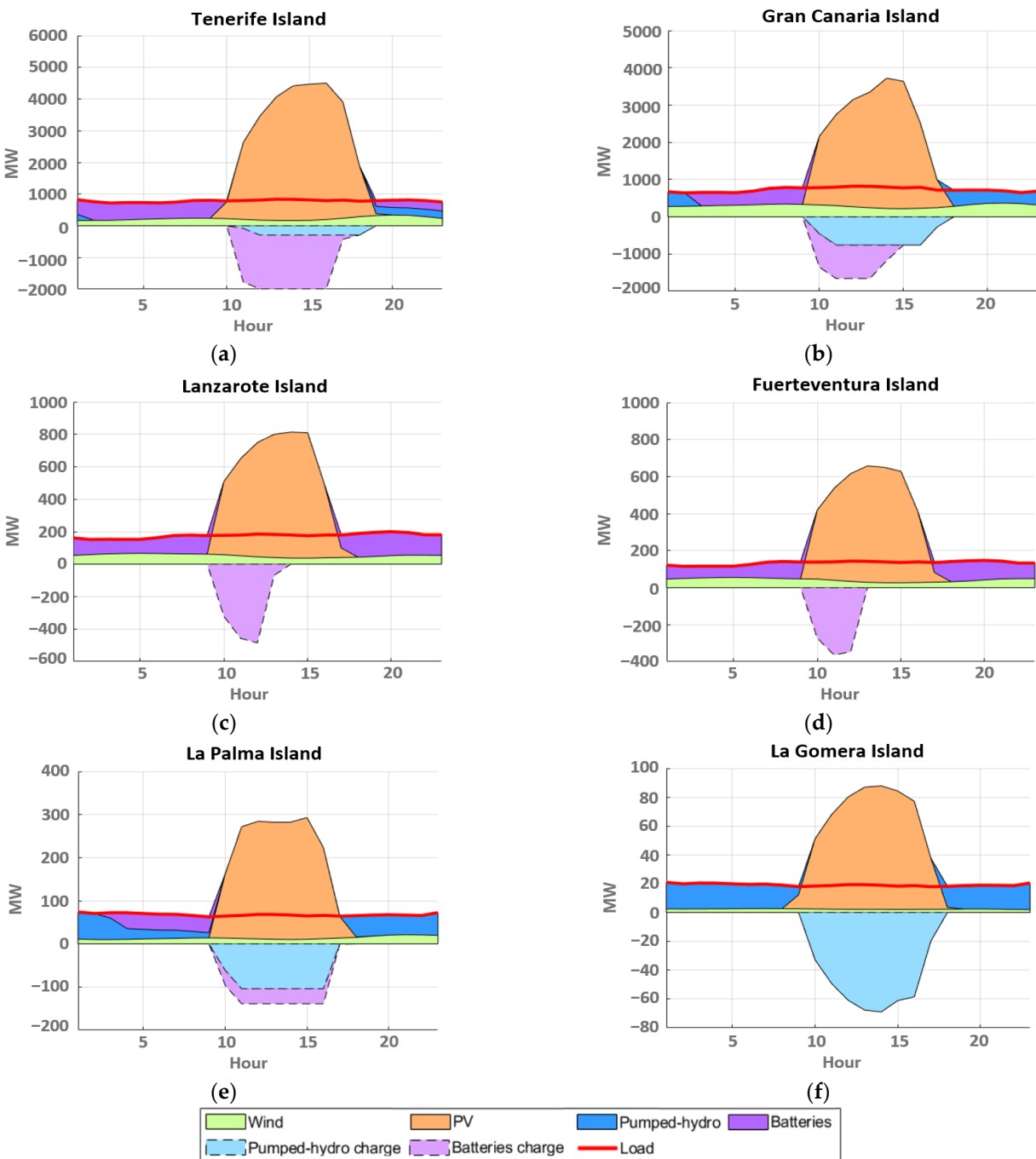

**Figure 16.** Hourly performance of the generation and storage systems on a typical winter day. (**a**) Tenerife; (**b**) Gran Canaria; (**c**) Lanzarote; (**d**) Fuerteventura; (**e**) La Palma; (**f**) La Gomera.

5.1.2. The Interconnected System

The interconnected system consists of the inter-island network connections shown in Figure 6. In the simulations carried out for this scenario, the cost of the interconnections has been considered, as well as the losses caused by this inter-island transport. The figures of both of these considerations are shown in Table 2.

Regarding the differences found between this interconnected system and the isolated generation systems of each of the islands, one advantage of the proposed interconnections is that, as mentioned above, at certain times the rest of the islands are able to compensate for the lack of energy from other islands. However, the final solution does not appreciably reduce the total generation power required, nor does it reduce the storage capacity needed

to cover the demand. As can be seen in Table 9, what it does accomplish is to redistribute generation, so that the installed capacity of solar PV increases, decreasing that of wind (which, as will be seen in the economic analyses, contributes to reducing the cost of generation, given the appreciably lower cost per MW of the installed capacity of solar PV technology in relation to wind), as well as that of biomass (although this contribution is small in both scenarios). In addition, there are no appreciable changes in the power and storage capacities of pumped storage facilities or mega-batteries (Table 9). However, there is a greater contribution from the storage of both technologies (Table 10), i.e., it is capable of absorbing a greater part of the excesses, so that these change from 33.8% in the non-interconnected system to 32.4% in the interconnected system. In other words, interconnections help to use an appreciable part of the energy surpluses produced on one island to satisfy the load demand on other islands and/or to charge their storage systems. Similarly, energy shortages can be covered by the generation from other islands or by their stored energy. One aspect to consider is that there is a variation in the load curve to be covered; in the previous scenario it was the individual load curve of each island while now it is the aggregate of all the islands, plus the losses associated with the interconnections, which can also influence the determination of the optimal mix. All this leads to a better performance of the system with interconnected islands, on the one hand, because of its lower installed power, but also because it is a larger system, which gives it advantages, for example, in terms of supply reliability and grid stability. In addition, as will be detailed in the section on economic results, there is a significant reduction in the cost per kWh produced.

**Table 9.** Comparison of the installed generation and storage powers and the storage capacity between the three scenarios analyzed.

| | Not-Interconnected | Interconnected | Optimized |
|---|---|---|---|
| | **Renewable sources** | | |
| **Solar PV (MW)** | 10,756 | 11,233 | 11,000 |
| **Wind (MW)** | 1478 | 1215 | 390 |
| **Biomass (MW)** | 366 | 100 | 150 |
| **Total power (MW)** | 12,600 | 12,548 | 11,540 |
| | **Storage systems** | | |
| **Hydro pump (MW)** | 1010 | 1010 | 2065 |
| **Battery system power (MW)** | 3663 | 3500 | 3500 |
| **Total power (MW)** | 4673 | 4510 | 5565 |
| **Pumped storage capacity (GWh)** | 16,636 | 16,636 | 16,636 |
| **Battery system capacity (GWh)** | 36,282 | 34,672 | 34,672 |
| **Total storage capacity (GWh)** | 52,918 | 51,308 | 51,308 |

**Table 10.** Comparison of the electric generation between the three scenarios analyzed.

| | Not-Interconnected | Interconnected | Optimized |
|---|---|---|---|
| | **Renewable sources** | | |
| **Solar PV (GWh)** | 25,030 | 26,692 | 26,138 |
| **Wind (GWh)** | 6767 | 5463 | 1753 |
| **Biomass (GWh)** | 183 | 17 | 2 |
| **Total power (GWh)** | 31,980 | 32,172 | 27,893 |
| | **Storage systems** [1] | | |
| **Hydro pump (GWh)** | 2307 | 2707 | 4801 |
| **Battery (GWh)** | 3242 | 4360 | 4551 |
| **Total energy (GWh)** | 5549 | 7067 | 9352 |
| | **System excesses** | | |
| **Surpluses (%)** | 33.8 | 32.4 | 19.1 |

[1] Energy re-fed to the grid by the storage systems; this energy comes from renewable generation.

As for the general performance of the interconnected system, it is quite similar to that shown previously for the island of Tenerife (Figures 10–12), so what was discussed in the previous section is applicable here. It must, however, be emphasized that, as shown in Figure 17a, the pumping system is clearly not capable of providing the necessary energy in periods of energy deficit. The system is empty every day of each month, except January; this is because in the initial calculation of the scenario (1st of January) the system is supposed to start from 50% of the water volume in the upper reservoir but, in any case, the monthly minimum is also zero. Thus, it has become essential to use an alternative system which, as in the rest of the study, has been reported to the mega-batteries (Figure 17b), for which it can be seen that in the months of January, October, and December there are moments with a total discharge. In the summer months, however, they are practically fully charged.

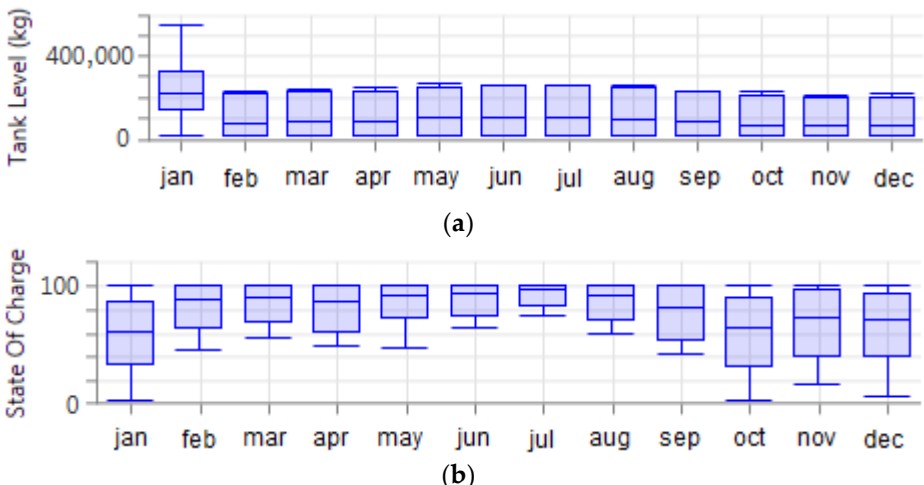

**Figure 17.** Monthly data of the storage systems. (**a**) Reverse pumping; (**b**) mega-batteries.

Additionally, as shown in the hourly representation of typical summer and winter days (Figure 18), these periods are the least and most adverse situations for the system, respectively. As for the previous case, since the greatest generation contribution comes from solar PV, then the lower radiation and sun hours are in winter months, so this is the worst scenario. In both cases, two storage system are needed, but in the typical winter day (Figure 18b) the pumping subsystem of the reverse storage is only able to cover the first night hours, after which the discharge of the mega-batteries provides the demanded energy. When electrical surpluses begin, both subsystems start their replenishment. However, the upper dams of the pumping system cannot be completely refilled (indeed, their capacity is less than 50%), because at maximum pumping power they need about 16 h for to refill completely, and in the winter months the hours with excess production in the system barely reach 7 or 8. In the summer months (Figure 18a) the situation is similar but with a greater refilling of the two systems, although in the case of pumping limited by the high refilling time (Figure 17a), while the batteries are close to their maximum charge (Figure 17b). Therefore, in spite of the abovementioned issues, the pumping system is able to provide the necessary energy until later in the night, so that the battery input is delayed, with the consequent lower discharge of the batteries.

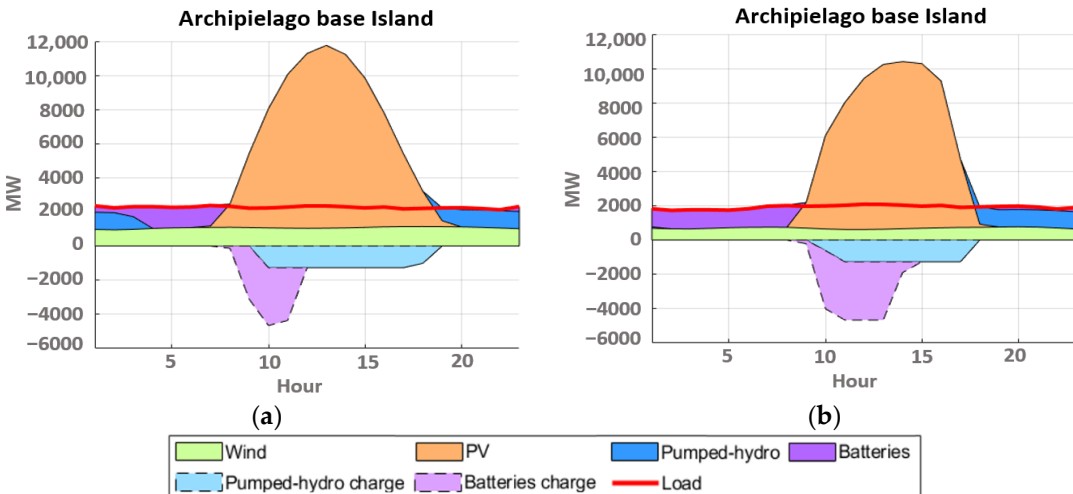

**Figure 18.** Hourly performance of the generation and storage systems in a typical day of (**a**) summer and (**b**) winter.

### 5.1.3. Optimized Interconnected System

After the description of the previous interconnected system, a series of tests have been carried out to optimize the characteristics/limitations of the facilities being considered up to this point. For example, as summarized in Table 6, there are different stored powers and energies for the pumping installations of each island from the archipelago which offer the possibility of installing reverse pumping. However, the energy to be stored is a condition given by the constraining circumstances existing on each island (in fact, in the proposed facilities no new dams are built, but the existing ones are used, so this cannot be increased unless new ones are built). Nonetheless, the pumping/turbine power is a condition that can be analyzed, in this case, according to the report of the Canary Islands Government [43] and, depending on the island, the installation can work for between 12 to 20 h at its maximum capacity for absorbing or transferring energy to the network. However, as shown in Figure 19 (example of an application to the Gran Canaria Island with 16 h of work at a full load), the pumping installations have a reduced operation capacity mainly throughout the winter, since the upper dams are almost emptied all the time. On the other hand, during the summer periods, they are almost filled all the time. This disadvantage is due to the circumstance that most of the electricity used by the pumping installations comes from solar PV surpluses, which are much lower in winter, when there is usually bad weather and the nights are also much longer. Consequently, there is an energy deficit, since during the summer night hours the system is not able to return all the energy stored during the day, while the opposite situation occurs during the winter, as during the daytime hours the system is not able to recover up to its maximum capacity. To solve this problem, the pumping/turbinating power has been increased, maximizing the pumping capability to take profit from the few hours of sunlight in winter, reaching the maximum storage capacity, while maximizing the turbinating capability to return to the grid all the energy stored during the day in the reduced hours of the night in the summertime. This change leads to a very significant increase in the energy surpluses that the pumping storage system can absorb and re-feed to the grid.

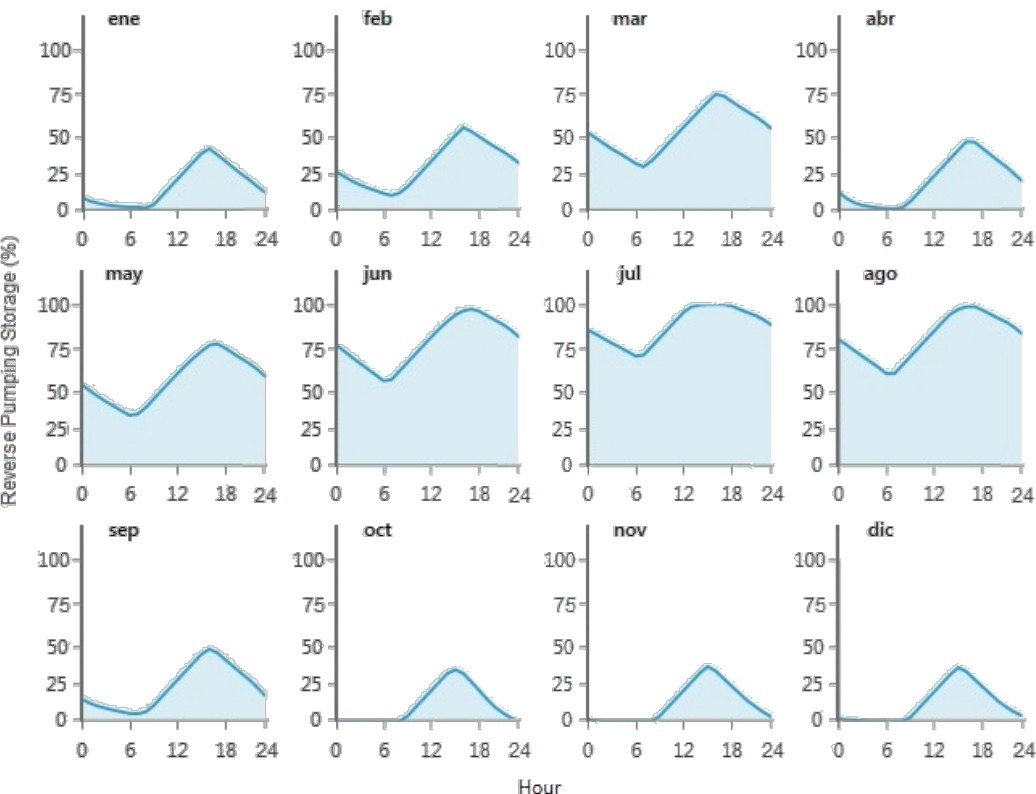

**Figure 19.** Hourly values of the monthly averaged storage capacity in the reversible pumping system for the Gran Canaria Island.

In addition, it also has been seen in previous analyses that the proposed mix uses the maximum installable capacity of solar PV, so an increase in this capacity should be considered if possible. In fact, according to the Deloitte Monitor report [25] about 10–11 GW of renewable installed generation capacity will be needed in the whole archipelago, from which 80% corresponds to solar (self-consumption, utility-scale), while the remaining 20% comes from wind energy. For this purpose, estimates have been made of the areas available for the installation of solar panels and, given the high occupancy assumed for self-consumption (70% of the total available rooftops), it has been considered unrealistic to increase this to a greater extent. However, as mentioned above, there is the possibility of seeking possible sites for solar farms or using the surfaces of the large number of reservoirs on the islands (many of them are used in the reverse pumping facilities) to generate further power. Therefore, in addition to the maximum power that can be installed for self-consumption, the possibility of an extra installation of solar PV power has been analyzed. This option has been explored, but no significant extra solar PV power has been needed.

In fact, if the installed power is considered, the proposed system consists almost entirely of solar PV generation (slightly over 95%), with wind generation reduced to just under 4% and biomass to just over 1%. This low contribution of wind power is partly due to the high generation costs of off-shore technology (selected because of the inconveniences of on-shore technology, mainly due to environmental and tourism issues for the installation of large wind farms on all of the islands). In any case, with an adequate storage system, such as the one proposed in this document, the weight of solar generation will always be dominant. It should also be noted that the contribution of biomass is testimonial, but that a slight increase in the installed power could have been considered, up to around 400–500 MW of installed power, given that in this way all the islands' resources could be used, and a backup system would also be available in case of need in exceptional circumstances.

Regarding the results of the scenario in comparison with the two previous ones, there has been an appreciable decrease in the necessary generation power. This has been due to the reduction in generation surpluses, which have gone from more than 30% in the first two scenarios to levels below 20%. This is because the increase in the power of the storage facilities (in reality, only the pumping power has been increased) leads to the excesses of the central day hours to be absorbed to a much greater extent. Thus, compared to the non-optimized interconnected scenario, the energy produced has been reduced by almost 20%, from more than 32 TWh per year to less than 28, while the energy re-fed into the grid by the storage systems has grown by more than 30%, from 7 TWh per year to more than 9 TWh per year.

Biomass has a testimonial role if the energy generated is considered. However, this generation resource is important mainly because of its role as a reliable and fast-activating generation source, so this technology is the only one of the three sources of generation suitable to be used as a backup in case of need. The electrical generation yields implemented in the program are those of gasification plus gas turbine, i.e., syngas production, with an efficiency of around 70%, and gas turbine which is about 30% [59]. In addition, as mentioned above, biomass is suitable for using the internal biomass resources existing on the islands, which otherwise would probably be simply burned in situ at the points of felling, pruning, etc. As such, the electric generation capacity of the islands with their own biomass resources is about 50 GWh per year (with the abovementioned gasifier plus gas turbine technology), with islands having to import any amount exceeding this quantity so that the biomass plants are ready to be activated if necessary.

The general performance of the optimized interconnected system is similar to that described for the previous scenarios (and summarized in Figures 10–12). Yet it must be emphasized that, as shown in Figure 20a, in the previous scenario the pumping system is not capable of providing the necessary energy in periods of energy deficit, but nonetheless has a much better performance. This is because the system is empty every day of each month but, in this case, due to its major pumping/turbinating powers, the system is capable of recovering/re-feeding during the day/night cycles. The mega-battery system (Figure 20b) continues being required, but in this scenario it is only during the month of December that total discharge of the system takes place. Indeed, the mega-battery system is hardly used and remains practically fully charged in the summer months.

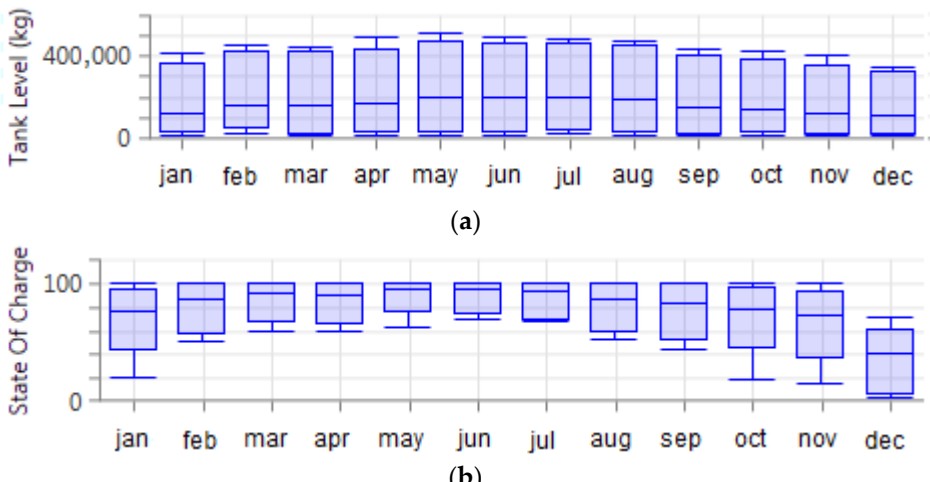

**Figure 20.** Monthly data of the storage systems. (**a**) Reverse pumping; (**b**) mega-batteries.

The hourly representation of typical summer and winter days is displayed in Figure 21. As for the two previous cases, given that the greatest contribution to electricity generation comes from solar PV energy and, as the lowest radiation and sunshine hours occur in the winter months, then this is the most critical moment. In both cases, the two storage

systems are needed, but on a typical winter day (Figure 21b), the turbinating subsystem of the reverse storage can cover a smaller part of the demanded energy during the night than in a typical summer day and, consequently, the mega-battery system must provide this extra contribution. Sometime after sunrise, when the electrical surplus begins, both subsystems start their replenishment, both in summer and winter, but in typical winter days there are not enough sun hours to completely fill the storage systems. In this scenario, in typical winter days, the upper dams of the pumping system are not completely filled, but at a higher capacity than in the previous scenario (capacity of at least 70% in the worst situations, Figure 20a), given that in this case at maximum pumping power the systems is refilled in about 8 h. Similar situation takes place for the battery system (Figure 21b), particularly in December, when the worst scenario occurs, since the monthly average charge is barely 40%. In the summer months (Figure 21a) the situation is similar, but usually with the complete refilling of the two storage systems. Consequently, the pumping system can provide a higher percentage of the demanded energy into the night, so that the entry of the mega-battery system is delayed, resulting in lower discharges.

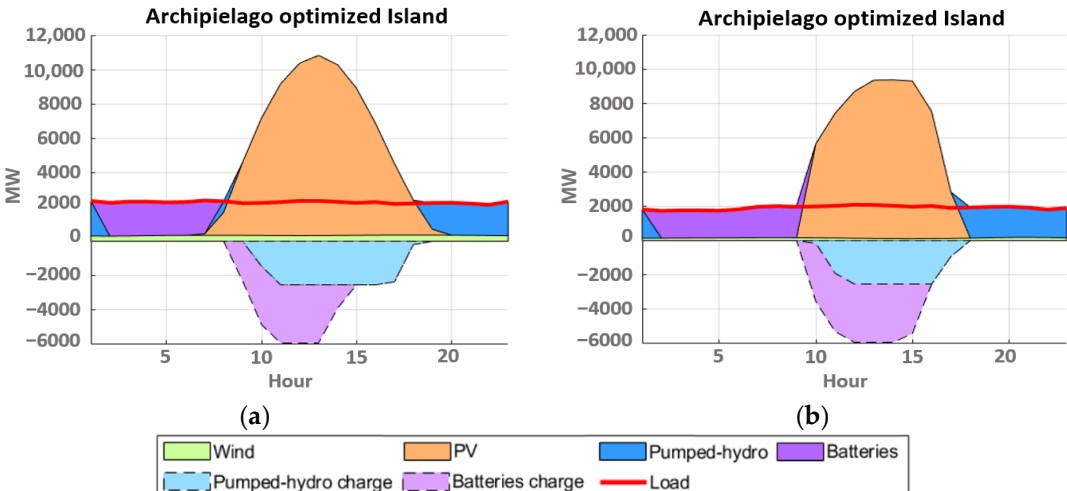

**Figure 21.** Hourly performance of generation and storage systems in a typical day of (**a**) summer and (**b**) winter.

5.1.4. Scenarios Comparison

The comparison of the power and storage capacities required from the different generation sources and storage systems for the three scenarios shows that interconnection provides a greater energy management capacity (Table 9). That is to say, at specific times the shortages and/or energy surpluses in some islands can be compensated by the others, so that the power to be installed in generation systems and the power and storage capacity required are slightly reduced. This power reduction in the generation systems is accentuated in the optimized scenario, although it is compensated for by the greater power of the reversible pumped storage system which is considered here (the storage capacity is conserved, since it is limited by the characteristics of the existing locations). In all of the scenarios, biomass generation represents a residual value but, as mentioned, its existence can be very important, and even its maximization given its reliability, so that it can be used as a backup under an exceptional situation, either due to a system failure or due to a specific lack of a resource, i.e., wind or solar. It should also be added that in the generation there is a redistribution of the powers of the generation systems, highlighting that the wind generation power to be installed is fundamentally reduced, increasing the solar power. This variation leads to higher generation peaks in the central hours, which the system of the second scenario is capable of absorbing to a greater extent than that of the first scenario, precisely because of the aforementioned interconnection. However, as already mentioned in the economic section, the cost of solar production is lower, which makes the

whole system cheaper. In this second scenario, the power to be installed for storage is also reduced; although it remains constant between the second and third scenarios, the reduction occurs in the batteries since, for economic reasons, the entire pumping installation capacity existing on the islands in the three scenarios is used.

With respect to energy production (Table 10), the second scenario is the one that produces more energy generation to cover demand, due to the fact that the interconnection system has associated losses, together with those of the storage systems. As can be seen in the table, the total energy stored has increased from 5.5 to 7.1 TWh per year, i.e., the interconnection system has a greater capacity to absorb occasional energy excesses. Therefore, as mentioned above, the size of the solar generation system has increased to the detriment of wind generation, but with a greater increase in both power and generation of the system. Although the end result is that there are greater losses, at the same time, there is a greater use of energy (less energy waste, from 33.8% to 32.4% between scenarios 1 and 2), so that from the economic point of view the system is more competitive. This situation of increased energy passing through the storage systems is accentuated in scenario 3, reaching almost 9.4 TWh per year. However, in this case, it is also accompanied by a considerable reduction in the energy generated, precisely because of this greater capacity to absorb the generation peaks, especially solar generation in the central hours of the day. The final result is a scenario with energy wastage much lower than the other two scenarios, from 32–33% in the first two scenarios to less than 20% in the third one. This situation is also accompanied by a significant reduction in system costs, both total and operational, and maintenance costs. As a final summary of the economic aspects, total investment over the lifetime of the projects has fallen from more than 34,000 M EUR in scenario 1 to almost 29,000 M EUR in scenario 2, and then to less than 27,000 M EUR in scenario 3, while maintenance has also been reduced appreciably, from almost 18,000 M EUR in the first scenario to 13,500 M EUR in scenario 2 and to 6000 M EUR in scenario 3.

To conclude this section of the comparison of the different systems for the three scenarios, the performance of each of the generation and storage sources will be separately analyzed. In relation to the solar PV generation subsystems (Table 11), it can be said that in all the scenarios there are high capacity factors (CFs), since in all of them they are around 27%, and they can be considered high if they are approximately above 20%. This because they have an equivalent operating time of almost 2500 h per year, well above 2000, which is already considered a high value. Something similar occurs with the wind generation system (Table 12), which presents CFs above 50%, while the usual values are in the 20–40% range. These percentages represent equivalent times of around 5000 h per year, also well above the approximately 2500 h per year typical of onshore production. Both situations are due to the aforementioned ideal location of the Canary Islands, which makes it an optimal place for the exploitation of both generation sources. In both systems there is an increase in the CF; in the case of the solar PV system, this may be due to the greater flexibility to absorb surpluses from the interconnected grid scenarios. Meanwhile, for wind generation, the significant increase in the CF can be attributed to the better location of the wind farms of the interconnected systems (in the non-interconnected system, each island has its own facilities in its vicinity).

**Table 11.** PV system summary.

|  | Not-Interconnected | Interconnected | Optimized |
|---|---|---|---|
| **Rated power (MW)** | 10,756 | 11,233 | 11,000 |
| **Expected life (yr)** | 25 | 25 | 25 |
| **Mean output (MW)** | 2857 | 3047 | 2984 |
| **Mean output (MWh/day)** | 68,575 | 73,129 | 71,611 |
| **Capacity factor (%)** | 26.6 | 27.1 | 27.1 |
| **Total production (GWh/yr)** | 25,030 | 26,692 | 26,138 |
| **Hours of operation (hr/yr)** | 2327 | 2376 | 2376 |

**Table 12.** Wind system summary.

|  | Not-Interconnected | Interconnected | Optimized |
|---|---|---|---|
| **Rated power (MW)** | 1478 | 1010 | 390 |
| **Expected life (yr)** | 25 | 25 | 25 |
| **Mean output (MW)** | 772 | 624 | 242 |
| **Mean output (MWh/day)** | 18,540 | 14,967 | 5803 |
| **Capacity factor (%)** | 52.3 | 61.8 | 62.0 |
| **Total production (GWh/yr)** | 6767 | 5463 | 1753 |
| **Hours of operation (hr/yr)** | 4579 | 5409 | 4495 |

For the storage systems (Tables 13 and 14), it should be noted that in all cases the scenarios use the maximum reversible pumping power that can be installed in the archipelago (as mentioned in the economic analysis, this is due to the lower cost per power unit installed compared to the mega-battery system). The significant increase in the use of pumping is also notable if the non-interconnected system is compared with the interconnected system and the latter with the optimized one. The first increase in the capacity factor of the reversible pumping installation (CF changes from 13.9 to 16.3%) is due to the greater versatility of the interconnected network. In the second increase (CF moves from 16.3 to 28.9%), the greater versatility of the interconnection is combined with, above all, the greater installed power of the pumping installations, which significantly increases the energy that the system is capable of absorbing. In the case of the batteries, there is a much smaller increase, with appreciable differences between the non-interconnected scenario and the two interconnected scenarios (FC changes from 9 to 13% approximately), which can also be attributed to the greater flexibility of the grid due to its interconnection between the islands.

**Table 13.** Reverse pumping storage system summary.

|  | Not-Interconnected | Interconnected | Optimized |
|---|---|---|---|
| **Rated power (MW)** | 1010 | 1010 | 2065 |
| **Rated capacity (GWh)** | 16,636 | 16,636 | 16,636 |
| **Round-trip system efficiency (%)** | 80 | 80 | 80 |
| **Expected life (yr)** | 50 | 50 | 50 |
| **Capacity factor (%)** | 13.9 | 16.3 | 28.9 |
| **Electrical consumption (GWh/year)** | 2884 | 3384 | 6001 |
| **Electrical grid re-feed (GWh/year)** | 2307 | 2707 | 4801 |
| **Mean electrical output (MW)** | 263.3 | 309.0 | 548.1 |

**Table 14.** Mega-batteries storage system summary.

|  | Not-Interconnected | Interconnected | Optimized |
|---|---|---|---|
| **Number of batteries** | 14,652 | 14,000 | 14,000 |
| **Rated power (MW)** | 3663 | 3500 | 3500 |
| **Nominal capacity (GWh)** | 36,282 | 34,672 | 34,672 |
| **Round-trip system efficiency (%)** | 70 | 70 | 70 |
| **Autonomy (hr)** | 10 | 10 | 10 |
| **Expected life (yr)** | 25 | 25 | 25 |
| **Capacity factor (%)** | 8.94 | 12.58 | 13.13 |
| **Energy in (GWh/yr)** | 4631 | 6229 | 6501 |
| **Energy out (GWh/yr)** | 3242 | 4360 | 4551 |

As a summary of the three scenarios, the interconnection of the different grids of each island to form a larger one leads to the greater versatility of the whole system. This versatility has an impact on a greater use of the storage systems (in the third scenario, the greater installed power of the pumping system has an important influence, which significantly increases its FC, although the storage capacity of the system does not change).

Thus, with the same electricity production from solar PV generation, it is possible to recover a greater amount of this energy through the storage systems. This increased use of storage systems leads to a significant reduction in wind power generation (as found in the economic analysis due to its higher costs) while maintaining as constant the installed power of solar PV.

*5.2. Economic Analysis of the Systems*

An economic analysis summary is displayed in Table 15. By analyzing this table, it is shown that the total costs of the systems reduce significantly from slightly below EUR 60/50/40 thousand million for the non-interconnected/interconnected/optimized systems, respectively. The cost of the non-interconnected system is nearly 50% higher compared to the interconnected and optimized system during the whole life of the systems. Concentrating on the initial capital needed to deploy the systems required in each scenario, it can be seen that this also changes appreciably, with reductions of around 15% and 10%, respectively, between the three scenarios (34,172, 29,523, and 27,383 M EUR, respectively). The replacement costs of all three systems are quite similar, as the solar PV system makes the major contribution, which represents most of the generation systems in all three scenarios and, consequently, has similar contributions. The same comment is not true for the operating and maintenance, where wind turbine O&M costs are much higher than the rest. Thus, given that the first scenario requires more wind power generation than the second, and that the second scenario requires more than the third scenario, then the cost reduces appreciably as the systems improve.

In order to carry out the different economic analyses, a lifetime of the joint systems (all three scenarios) of 50 years has been considered. As the lifetime of the pumping system is 50 years, while the rest of the subsystems last for 25 years, i.e., the PV, the wind system, and the storage system, then, it has been taken as a minimum common multiplot of 50 years, which would imply that one replacement of all systems (except for the pumping system) after 25 years of the project has been considered.

**Table 15.** Initial capital, replacement, O&M, and total discounted costs for the three generation scenarios.

| Scenarios | Systems | Capital (M EUR) | Replacement (M EUR) | O&M (M EUR) | Total (M EUR) |
|---|---|---|---|---|---|
| Non-interconnected | PV | 16,436 | 4059 | 1184 | 21,678 |
| | Wind turbine | 3247 | 968 | 13,656 | 17,871 |
| | Pumped storage | 3150 | 0 | 33 | 3183 |
| | Battery system | 11,339 | 1605 | 2850 | 15,794 |
| | Total | 34,172 | 6632 | 17,723 | 58,526 |
| Interconnected | PV | 15,650 | 4006 | 856 | 20,512 |
| | Wind turbine | 2730 | 609 | 11,000 | 14,339 |
| | Pumped storage | 2230 | 0 | 24 | 2254 |
| | Battery system | 6440 | 1530 | 1615 | 9585 |
| | Interconnection | 2473 | 0 | 258 | 2731 |
| | Total | 29,523 | 6145 | 13,753 | 49,421 |
| Optimized | PV | 15,350 | 3906 | 840 | 20,096 |
| | Wind turbine | 880 | 195 | 3510 | 4585 |
| | Pumped storage | 2240 | 0 | 24 | 2264 |
| | Battery system | 6440 | 1530 | 1615 | 9585 |
| | Interconnection | 2473 | 0 | 258 | 2731 |
| | Total | 27,383 | 5631 | 6247 | 39,261 |

An economic analysis summary is shown in Table 16. It is extremely relevant to note that in any scenario, even with large initial investments to implement the different systems, the return-on-investment periods are usually relatively low, especially if future electricity costs evolve in the same direction as in recent months. Aiming to check the effect

of the reference costs of the electric generation on the system's performance, conservative and volatile scenarios have been displayed in the table, so a variation range for the involved variables is provided. This last situation towards high generation costs is highly probable given the necessary change in the current energy mix to clean energy generation systems, which will make it possible to dispense with fossil fuels in their entirety, foreseeably by the year 2050 at least in Europe; these are aspects that will undoubtedly raise the costs of energy generation. It must also be noted that solar PV generation is much more economical than offshore wind generation, while reverse pumped storage is also much more economical than battery storage. However, in the end, it is worth highlighting that the LCOE of the optimized system is almost 10 cEUR/kWh, which is a very tight cost for this type of systems and for the size of the one designed in this document. We must also mention the high return on investment and internal rate of return for all three systems, especially the optimized one, which even reaches double digits in the internal rate of return. We conclude this section by highlighting the high performance of all the scenarios, since all the proposed scenarios can be considered viable and profitable. The segmented values by islands are displayed in Table 17, which shows that all the islands have lower system costs than the conservative ones, except for the non-interconnected scenario in the island of Tenerife, a situation motivated by its high energy demand and high ratio compared to its pumping storage capacity (which leads to the need to resort to a large capacity of mega-batteries with the consequent cost overruns). Not even the islands of Lanzarote and Fuerteventura, which do not have pumped storage, reach such high values, having significantly lower values compared to the conservative generation costs, around 14 compared to the more than EUR 15 cEUR/kWh. Moreover, La Gomera has a high pumped storage capacity and low consumption. However, all the islands have significantly lower values than the volatile case.

**Table 16.** Initial capital, replacement, O&M, and total costs for the three generation systems.

|  | **Non-Interconnected** | **Interconnected** | **Optimized** |
|---|---|---|---|
| LCOE solar PV (cEUR/kWh) | 3.5 | 3.2 | 3.2 |
| LCOE wind (cEUR/kWh) | 15.5 | 12 | 12 |
| Reverse pumping cost (cEUR/kWh) | 2.8 | 1.7 | 0.9 |
| Battery wear cost (cEUR/kWh) | 9.7 | 4.4 | 4.2 |
| System LCOE (cEUR/kWh) | 15 | 12.4 | 10.1 |
| Return on investment * (%) | 2.7–6.1 | 4.3–7.9 | 5.8–9.5 |
| Internal rate of return * (%) | 4.2–8.4 | 6.3–10.4 | 8.1–12.2 |
| Simple payback * (yrs) | 18.0–11.0 | 13.5–9.1 | 11.2–7.9 |

* Two comparative scenarios have been analyzed versus the proposed systems, one conservative (generation costs of the Canary Islands of 2019) and one volatile (costs of the current year), so a variation range of the different variables is provided.

**Table 17.** Economic analysis summary of the non-interconnected scenario segmented by islands.

| Island | LCOE (cEUR/kWh) | Return on Investment (%) | | Internal Rate of Return (%) | | Simple Payback (yrs) | |
|---|---|---|---|---|---|---|---|
|  |  | **C** | **V** | **C** | **V** | **C** | **V** |
| Tenerife | 16.1 | 2.0 | 5.1 | 3.3 | 7.2 | 20.2 | 12.2 |
| Gran Canaria | 14.2 | 3.2 | 6.9 | 4.9 | 9.3 | 16.5 | 10.1 |
| Lanzarote | 14.1 | 3.2 | 6.8 | 5.0 | 9.4 | 16.1 | 9.8 |
| Fuerteventura | 14.4 | 3.0 | 6.5 | 4.7 | 9.0 | 16.6 | 10.1 |
| La Palma | 14 | 3.3 | 7.0 | 5.0 | 9.3 | 16.6 | 10.2 |
| La Gomera | 10.3 | 5.7 | 9.4 | 7.8 | 11.7 | 12.0 | 8.3 |

## 6. Conclusions

This last section of conclusions summarizes the most important results of the analysis conducted in this study, as follows:

-   The interconnection of the system has led to an increase in the optimal weight achieved by the solar PV generation system. This increase is motivated by economic reasons, given the lower cost per kWh of this technology compared to wind energy (3.5 compared to 15.5 cEUR/kWh of the solar PV and offshore wind systems). However, the increase in solar generation produces a higher peak in the central hours of the day, which can be partially absorbed by increasing the storage capacity with the corresponding cost overrun. Or, as in this case, with the interconnection between islands, it has been possible to maximize above all the existing storage capacity, so that at specific times some islands have been able to absorb to a greater extent the excesses of others of them (a strong increase in energy passing through the storage systems). The final result is a system with greater flexibility;
-   The optimization of the system in scenario 3, especially the maximization of the installed power of reversible pumping, has led to an even greater use of the storage systems. As a result, the generation needs of the system have been reduced, both in solar generation and, above all, in wind generation. The final figures show a reduction of approximately 13% in the energy generated by the system;
-   This high capacity to absorb the excess energy by the storage systems in scenario 3 has led to low energy waste values (approximately 19%). This is even lower considering that the implemented system is entirely based on renewable energy generation, for a scenario of total electrification of the economy, in a system of a considerable size (a demand of approximately 18 TWh/year) and being isolated from another external grid;
-   The LCOE of the proposed mix, especially in the case of the last scenario, is very low for a system of the proposed characteristics, being about 10 cEUR/kWh. This makes it a competitive system with a reduced payback time (less than 6 years for the volatile case) for a totally autonomous and off-grid system;
-   As can be deduced from the different analyses developed throughout the document, both the economic and technical figures show improvements among the different scenarios, i.e., there are progressive advances when passing from the non-interconnected scenario through the interconnected to the optimized one. In this sense, there is a reduction of approximately 10% in the installed power between the non-interconnected scenario and the interconnected and optimized scenario. These figures reduce to only 3% when analyzing the results of the storage capacity required to manage the system. For energy wastages, this variable reaches a reduction of almost 15% (excesses of almost 35% of the energy produced have gone down to 19% between the non-interconnected and the interconnected-optimized scenario). Finally, we reach the best results in the economic field, with a saving of almost 35% in the total costs of the installation with interconnections and the optimized system compared to the non-interconnected one;
-   It should also be noted that biomass, despite representing a residual percentage in all scenarios, is a generation technology that should continue to be present in the mix of any scenario selected. On the one hand, this technology is the optimal one to provide a backup in case of need, since it is free from the intrinsic variability of wind and solar resources. In addition, the consideration of this technology in the energy generation mix allows the use of forest resources that would otherwise remain in the forest or would simply be burned;
-   It has been observed that a reduced specific weight is necessary for offshore wind power generation if there is a high storage capacity that is able to manage a significant part of the inevitable excesses, especially in the central hours of the summer days, produced by solar PV generation;

- The power output of the storage systems has attained a value of around 50% of the peak power of the generation system. This high value is reached because the generation is based on solar energy, and probably lower values would be obtained if wind generation were used, given the high average and fairly constant values that are found in the islands, although in other locations this situation could vary appreciably;
- A similar situation to that described for the power of the storage systems is found for their storage capacity. The system is required to be able to absorb at its maximum power for almost 5 h. This value could vary appreciably if wind-based generation is considered or for a system deployed at another location;
- It has been shown that the use of batteries is necessary to manage the system, although they have been the last option once all possible pumping facilities on the islands have been used. Other alternatives could be explored instead of the use of mega-batteries for an installation with such high requirements as the one analyzed, although at present this is probably the only option. However, in the future, the possibility of using hydrogen as an energy vector could be analyzed, given its possibility of storage;
- The ratio between installed power and maximum consumption is approximately 5, which is a fairly low ratio. If the aim is to further reduce these values while maintaining zero greenhouse gas emissions, there are several options. The first and simplest would be the maximization of the systems with high reliability, such as the biomass system, to take advantage of the resources of the islands, and ultimately the importation if necessary. Another option would probably be its complementation with the utilization of gas turbine generation (ideally fueled by "green" hydrogen, although there are at present significant problems, particularly in terms of efficiency, in the transportation of hydrogen production and the costs of storage). Lastly, other possible alternatives, in this case not for backup but for base-load power, but also with no emissions of greenhouse gas, would be the installation of nuclear reactors (likely small modular reactors, or SMRs). However, this possibility is currently not feasible in Spain because to the presence of a nuclear moratorium that prohibits their installation, in combination with the usual objections, principally due to waste management and the possibility of accidental situations;
- The utilization of this surplus generation for other uses could be considered as a potential topic to be studied in subsequent analyses. For example, the production of hydrogen could be studied, an energy vector that would allow the subsequent use of these energy surpluses. Hydrogen could be useful for sectors that are challenging to electrify or to export to other places;
- The realization of sensitivity studies related to uncertainties in the future costs of generation and storage of the several technologies used is another possible option; although this is beyond our current objectives, it could be tackled in future work. Moreover, uncertainty analysis associated with the ability of the system to be self-sufficient due to the inherent variability/uncertainty associated with the renewable resources availability (solar and wind) is also necessary;
- The analysis of the potential impacts of potential generation and demand management policies may also be addressed for future research. In particular, this might include studying the impact of incentivizing the deployment of solar panels and/or prioritizing consumption at specific hours (to shift the demand curve towards production), among other possible measures.

To conclude, it should be noted that the methodology presented in this study can be adapted and applied to any autonomous or interconnected grid (setting the desired system coverage). For this, it should only be required to make the modifications to the wind and solar photovoltaic resources and, if necessary, to implement other potential sources to be considered, for which the available resources would have to be reintroduced.

**Author Contributions:** C.B.-E., Y.R.-D., Y.C.-C. and J.L.M.-C. Conceptualization, C.B.-E. and Y.R.-D.; methodology, C.B.-E., Y.R.-D., and J.L.M.-C.; software, Y.R.-D. and Y.C.-C.; validation, C.B.-E., Y.C.-C. and J.L.M.-C.; formal analysis, C.B.-E. and Y.R.-D.; investigation, C.B.-E. and Y.R.-D.; resources, Y.R.-D. and Y.C.-C.; data curation, Y.R.-D. and Y.C.-C.; writing—original draft preparation, C.B.-E. and Y.R.-D.; writing—review and editing, Y.C.-C. and J.L.M.-C.; visualization, Y.R.-D. and Y.C.-C.; supervision, C.B.-E. and J.L.M.-C.; project administration, C.B.-E.; funding acquisition, J.L.M.-C. All authors have read and agreed to the published version of the manuscript.

**Funding:** This research received no external funding.

**Data Availability Statement:** Not applicable.

**Acknowledgments:** The authors would like to express their gratitude to the Generalitat Valenciana (Spain) for its support under the Santiago Grisolía Program/2018/140. The authors would also like to extend their gratitude to the Ministerio de Economía, Industria y Competitividad and by Agencia Nacional de Investigación under the FPI grant BES-2017-080031.

**Conflicts of Interest:** The authors declare no conflict of interest.

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
