# Peer review of "Assessment of a Fully Renewable Generation System with Storage to Cost-Effectively Cover the Electricity Demand of Standalone Grids: The Case of the Canary Archipelago by 2040"

_machines, doi:10.3390/machines11010101_

Round 1

Reviewer 1 Report

Yago Rivera-Durán et al. have simulated a scenario with a totally renewable generation system and with total electrification of the economy taking the example of Canary Islands in 2040. They proposed an autonomous generation system with no CO2 emissions. Overall, this paper is with clear logic flows, and claims well supported by data. I would thus recommend acceptance of this work with the following revision suggestions.

1. In part 3.2, Scenarios of Interconnected Grid between Islands, 7 islands have many ways of connections, is the closest neighbor the optimized connection? Will the energy consumption and population affect the choice?

2. In part 4.4, biomass system is discussed. However, the data on this is mainly estimation. And from the discussion part, it’s mainly as a supplement source. What is the importance of this part to the result then? If by 2040, the biomass system cannot be commercialized to the estimated extent, will any conclusion of this paper change?

3. In part 5.2, the cost of initial installation and replacement are considered. But is the lifetime of the PV, wind system and the storage systems already taken into account?

4. On page 35, line 1065 and 1084, refs are shown as errors.

Reviewer 2 Report

Dear authors,

thank you for providing your interesting study on the renewable energy supply of the Canary Islands using electrical interconnections between the islands. For your selected scenario, you principally show that a renewable power supply is possible (using appropriate, but realistic infrastructure) and point out the technical and economic benefits of electrically interconnecting the islands.

From a higher-level perspective, I first offer the following comments:

The introduction is comprehensive and provides good understanding of the subject matter. Also, a sufficient number of literature references is provided throughout the entire article. The methodology is described in a more qualitative manner since the biggest fraction is done within the HOMER modelling tool. This is OK from my poitn of view since HOMER is described elsewhere (which you refer to). All assumptions are very detailed and are justified with the respective literature / data source. The results are presented to the readers in an understandable and comprehensible manner.

However, from my point of view, there is one key point that this study should fulfill so that the results will stand in reality:

I refer to page 11 (line 421) and page 12 (line 456):
The year 2019 (not being an “unusual year” as you write) was used as a reference for solar irradiation and wind production. However, with a high likelihood there also occur years with less sun and less wind. In order for the Canary Islands to have an entirely reliable renewable energy supply even in years with poor sun and poor wind, this study should not be based on a "not unusual" (i.e. a normal) year, but instead on a year with unfavorable weather conditions.
To give this study more realism, the basis of the weather data should either be (a) a poor wind and solar year, or (b) a multi-year period in which both good and bad years occur.

I would kindly ask you to use such data as a basis for this study.

If you see this differently, I gladly await your argumentation (which should then also be included in the article).

Comments / suggestions to specific passages:

Section 3: Methodology:

It should be mentioned in which time granularity the simulations have been carried out (if I have not overseen it).

This study considers the effects of interconnecting the different islands. However, it is unclear if the electric networks within the individual islands have been considered as well. It can be seen e.g. from Figs. 18 and 21 that the power levels of energy production and storage exceed the consumption power (for which the island networks are designed right now) by far. It is possible that this requires a massive extension of the electrical networks per island. Maybe, there are ways to avoid this (or maybe not). In order to provide a complete picture, the authors should address in how far intra-island networks have been included. If these networks have not been considered, this should be justified.

Page 8 / Figs. 3 & 4:

Are these 1-day profiles representative? Please justify briefly and mention the day/month of the shown profiles. Why are the respective power levels at 24:00 and 00:00 not identical? It might be furthermore interesting for the readers to see also the actual (i.e. today's) electric demand profiles of Lanzarote / the entire archipelago for a direct comparison.

Page 15 / Table 6; and reference 55:

Ref. 55 is not a classical battery storage but, more specific, a chemical redox-flow storage system. This should be mentioned. Please justify: Why did you chose this (more exotic) technology over well-established Lithium-ion battery systems? (considering that the scenario of your study is in the year 2040) 

Page 21+22 / Figure 13a+d and page 23, line 763:

As I understand, the pumped hydro storages on Tenerife and La Gomera need to be doubled (at least) just to cover the month January. The cost-benefit ratio for these storages is thus very poor, since in the remaining 11 months of the year, they are not even used to their half and should be commented. This can be used as an argument for the optimized interconnected system (Fig. 20a).

Figures 13 / 14 / 17 / 20:

The month names should be indicated in English.

Sections 5.1.2 & 5.1.3:

It would be very interesting (but not absolutely necessary) if the authors could show the mutual power exchange between the islands (e.g. in a time series), since this can be regarded as a measure for rebalancing the generations / consumptions between the islands, thus leading to energetic synergies and to a smaller dimensioned power generation system.

Section 5.1.4 (page 33; lines 1034 – 1042):

Summarizing, it can also be said that the amount of total energy production and storage capacities are in the optimized scenario only approx. 10% lower than in the not-interconnected scenario. This is of course a certain reduction, but it is not a fundamental improvement. This may be because the energy production and consumption profiles of the individual islands are quite similar which limit the amount of energetic synergy effects to be exploited with the interconnection. This should be commented in some way. However, a larger benefit of the interconnection comes in the economic perspective (Section 5.2) which can be referred to already here.

Page 35; lines 1065 & 1084:

broken references need to be fixed.
